# Generative Caching for Structurally Similar Prompts and Responses

**Sarthak Chakraborty**[1]*     **Suman Nath**[2]     **Xuchao Zhang**[2]     **Chetan Bansal**[2]
**Indranil Gupta**[1]

[1]University of Illinois at Urbana-Champaign, [2]Microsoft Research

`{sc134,indy}@illinois.edu, {sumann,xuchaozhang,chetanb}@microsoft.com`

## Abstract

Large Language Models (LLMs) are increasingly being used to plan, reason, and execute tasks across diverse scenarios. In use cases like repeatable workflows and agentic settings, prompts are often reused with minor variations while having a similar structure for recurring tasks. This opens up opportunities for caching. However, exact prompt matching fails on such structurally similar prompts, while semantic caching may produce incorrect responses by ignoring critical differences. To address this, we introduce `GenCache`, a generative cache that produces variation-aware responses for structurally similar prompts. `GenCache` identifies reusable response patterns across similar prompt structures and synthesizes customized outputs for new requests. We show that `GenCache` achieves 83% cache hit rate, while having minimal incorrect hits on datasets without prompt repetition. In agentic workflows, it improves cache hit rate by ∼20% and reduces end-to-end execution latency by ∼34% compared to standard prompt matching.

## 1   Introduction

AI applications, often composed of AI agents, augment Large Language Models (LLMs) with external tools [28, 40, 33], enabling them to interact with their environment and solve complex, multi-step tasks [45, 50, 41, 2, 31, 32, 59]. Many such applications repeatedly address similar task patterns, naturally leading to the construction of structurally similar prompts [21]. For instance, structured repetitive workflows like data entry or customer service agents encounter repetitive queries with minor variations, or cloud operations agents frequently diagnose recurring system faults [47, 38, 58]. Furthermore, prompt engineering techniques often format prompts with reusable templates (called *system messages* for AI agents), leading to similar prompts. Prior work has shown that caching and reusing prompts and their associated responses can substantially reduce latency and dollar cost in such repetitive workflows since the number of LLM calls is reduced [14, 63, 64, 43, 22, 21, 20, 57].

Existing client-side LLM caches are typically designed as key-value stores [7, 11, 12]. Traditional exact request (or prompt) matching returns a cached value if the new prompt exactly matches a stored prompt key [27, 39], while semantic caches return a cached response based on the semantic similarity of the new request with a stored prompt key [14, 20, 21]. Semantic similarity is computed using vector embeddings, where two keys are considered similar if their cosine distance surpasses a predefined threshold.

However, existing caching techniques fail in applications where prompt-response pairs exhibit three key properties: (a) **Structured Regularity:** prompts are not exact or semantically similar, but instead *structurally similar*, i.e. follow a consistent format (defined in §2), (b) **Controlled Variability:** prompts show minimal yet important controlled variations, and (c) **Predictable Response Pattern:** the responses also follow a predictable format, but differ across prompts, remaining *correlated* with the prompt variations. Consider a web-shopping agent [31, 59, 24] that queries an LLM to decide its next action from the page content and user instructions. Two example user instructions that satisfy

---

*Work done during internship at Microsoft Research.

39th Conference on Neural Information Processing Systems (NeurIPS 2025).

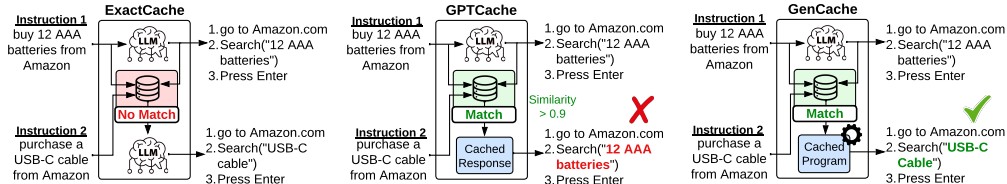

Figure 1: Comparison of `GenCache` with existing caching techniques. treats both instructions as distinct and results in cache misses, hence uses LLM to generate responses for both. GPTCache encounters cache hit for Instruction 2, but incorrectly returns the already saved response for a similar prompt. `GenCache` on the other hand, executes the cached program locally on cache hit to generate the correct response tailored to the input

these properties are: (1) *buy 12 AAA batteries from Amazon*, and (2) *buy a USB-C cable from Amazon*. Exact prompt matching (`ExactCache` in short) treats the two prompts as distinct, resulting in cache misses for both. Thus, it ensures correctness but reduces efficiency, since the prompts need to be identical for cache reuse. In contrast, a semantic caching method such as GPTCache [14] considers them semantically similar, producing a cache hit for instruction 2 but incorrectly reuses the same response from instruction 1, as shown in Figure 1. Thus, semantic caching improves cache reuse at the risk of erroneous results, which is potentially dangerous if agents perform non-reversible actions.

In this work, we propose *Generative Cache* (`GenCache` in short), a new client-side caching technique that balances the performance-correctness trade-off. Unlike traditional caches that return stored LLM responses verbatim on a cache hit, `GenCache` generates custom responses tailored to the prompt, achieving higher cache hit rates without compromising accuracy. This requires prompt-response pairs to satisfy the three key properties described above, common in AI agents and repetitive workflows. In the earlier example, the two instructions share a similar structure (verb-item-phrase) with minor variations, in verb synonyms (*buy* and *purchase*) and the item name. `GenCache` automatically discovers such structural similarity and generates a cache hit. On cache hit for instruction 2, `GenCache` *synthesizes* a custom response (similar in structure to the previous response) with the important variation in the item name. Figure 1 depicts this workflow.

`GenCache` clusters structurally similar prompt-response pairs based on their embedding similarity and generates a program that takes the prompts as input and synthesizes the correctly formatted responses as output. This program captures a common pattern of how the prompt maps to the response across all the prompt-response pairs in the cluster, and saves it as a cache. A new prompt request is matched with an existing cluster based on its embedding similarity, and the stored program is executed locally via a runtime like Python interpreter to generate a response. Unlike existing caches that store prompts and their LLM responses, `GenCache` stores prompts and the program that can generate these responses. Our evaluations on the Webshop dataset [53] demonstrate over 83% cache hit rate and at least 35% cost savings. On a synthetic dataset with higher structural similarity, cache hit rate is above 98%. When integrated with existing AI agents, it reduces the end-to-end execution latency and achieves 20% higher hit rate. In summary our contributions are:

- We propose a new caching technique, `GenCache`, for structurally similar prompts that can generate novel variation-aware responses for new prompts.
- `GenCache` identifies common patterns of generating response from structurally similar prompts within a cluster, and encodes the pattern in the form a program. `GenCache` then validates the program for correctness and stores it as cache.
- We compare our method with synthetic and benchmark datasets, and report our results by integrating it with two agent frameworks that perform repetitive tasks.

## 2 Problem Definition

Repetitive tasks or tasks following a similar template [62] are ideal for `GenCache` to extract the common pattern between multiple prompt-response pairs. Some common use cases are as follows.

**Key Use Cases:** Apart from the example in §1, other use cases are: (1) SRE agents [47, 38, 58], which execute documented remediation steps based on system alerts that follow some standardized templates. For example, alerts such as *"NSM to RNM connection is lost in useast1"* and *"NSM to RNM connection is lost in uswest1"* differ only by region, yet map to the same troubleshooting documentation, requiring identical actions by modifying the affected region. (2) Web agents managing

Google Maps often encounter recurring queries with slight variations, e.g., *"How long does it take to walk from Univ of Pittsburgh to starbucks on Craig Street?"* and *"... from Carnegie Mellon University to Univ of Pittsburgh?"*. These require generating similar API calls with different parameters. However, `GenCache` is less suitable for free-form chatbot interactions, which exhibit high prompt diversity, where semantic caching remains more effective.

`GenCache` is particularly effective for ReAct-style prompting [54] when the expected response is a structured action. In this work, we target use cases involving reversible actions, common across many applications like system fault identification (only read queries are issued to databases), adding items to a cart in an e-commerce website (can be removed if incorrect), checking distances between two endpoints in a map (endpoints can be rectified), etc. `GenCache` operates only on the final prompt constructed by an agent, which includes all contextual and user-specific personalization logic.

**Formal definition of Prompt Characteristics:** Let $x$ be an individual request (or, user instruction) from a set of repetitive requests $\mathcal{T}$. Let $\alpha(x)$ be the constructed prompt passed to the LLM by a client or an agent, and $\beta(x)$ be the corresponding response. Different characteristics of $\alpha(.)$ and $\beta(.)$ for two requests $x_1, x_2 \in \mathcal{T}$ provide distinct caching opportunities.

1. (Identical Prompts) If $\alpha(x_1) = \alpha(x_2)$ and $\beta(x_1) = \beta(x_2)$, it implies identical input prompts generate the same LLM responses, ideal for exact prompt matching.
2. (Semantically Similar Prompts) If $\alpha(x_1) \sim \alpha(x_2)$, but $\beta(x_1) = \beta(x_2)$ implies *Semantic Caching* techniques like GPTCache [14], InstCache [64] and others [20, 21] are ideal.
3. (Structurally Similar Prompts) If $\alpha(x) = a_1.x.a_2$ and $\beta(x) = b_1.f(x).b_2$ where $a_1, a_2, b_1, b_2$ are prompt phrases (e.g., phrases like *"buy"* and *"from Amazon"* in the prompt *"buy item X from Amazon"*) and $f(x)$ is a transformation on $x$ (e.g., extracting keywords like *"item X"* from the entire prompt using a regular expression) which are common over multiple requests, `GenCache` can automatically identify $a_1, a_2, b_1, b_2$ and $f(.)$ (may incorporate branching logic) from multiple examples using regex matching over $\alpha(x)$ and $\beta(x)$. This allows it to locally generate $\beta(x)$ for any new request $x$. Unlike `ExactCache` and *semantic caching*, `GenCache` does not require that $\beta(x_1) = \beta(x_2)$. Furthermore, since short prompts by nature exhibit only a handful of deviations in their way to write, `GenCache` can also capture such minor deviations (typically synonyms) in $a_1, a_2$ (for example, "buy" and "purchase"), by using an "OR" operator in the regex pattern.
4. (Structurally Dissimillar Prompts) When neither $\alpha(x)$, nor $\beta(x)$ show commonalities in their structure or phrasing, `GenCache` does not generalize well

## 3   `GenCache` **Design**

### 3.1   **Overview and Runtime Workflow**

The overall workflow of `GenCache` (Figure 2) follows the general caching paradigm, i.e., for each incoming input prompt $\mathcal{P}$, the system first identifies the relevant cache and attempts to reuse it. On a cache hit, the response is directly generated by the cached program; otherwise, the prompt is processed by the LLM as usual. Before returning the LLM-generated response, both the prompt and its output are stored in a database so that a cache generation attempt can be made. Cache generation involves–(I) cluster *structurally similar* prompts such that there is a consistent pattern in the way an LLM generates its responses, (II) infer the consistent pattern (we leverage an LLM) for each cluster and convert it to a program, and (III) validate the generated program before caching it for further use. For step II, we use in-context learning [15, 34] to help the LLM learn the mapping between prompts and responses within each cluster and generate a generalized program reflecting that relationship.

Along with the cached program, we store a regular expression that characterizes the structural pattern of prompts within its cluster, which aids in cache selection at runtime. For a new prompt $\mathcal{P}$, at runtime, the system first identifies the closest cluster based on cosine similarity between its embeddings to the cluster center. The stored regex then verifies whether $\mathcal{P}$ conforms to the structural pattern of that cluster. If validated and a cached program exists, `GenCache` executes it using a runtime like Python interpreter and returns the generated response after sanity checks. Cached programs include sufficient exception-handling blocks to capture cases where the program does not apply to $\mathcal{P}$.

### 3.2   **Prompt Clustering**

When a cache miss occurs, the LLM generates a response $\mathcal{R}$ for an input prompt $\mathcal{P}$. *Similar* sets of $(\mathcal{P}, \mathcal{R})$ pairs are clustered based on embedding similarity. Since prompts in LLM systems often

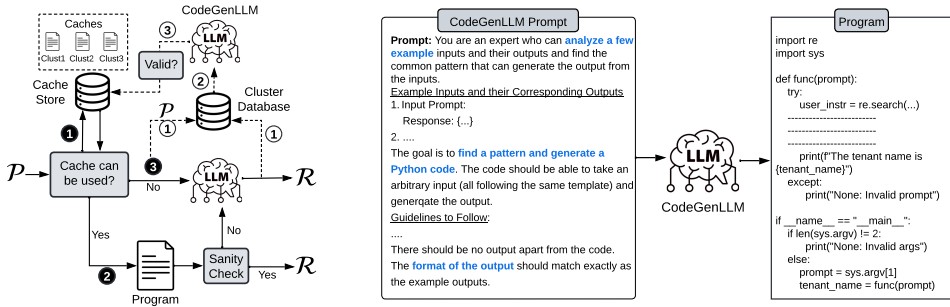

(a) Runtime Workflow       (b) Generated Program and its prompt

Figure 2: `GenCache` workflow (solid lines: cache reuse, dotted lines: cache generation). Figure 2a illustrates the runtime workflow–For a new prompt $\mathcal{P}$, the system finds the nearest cluster based on a similarity threshold and checks for an available cache for reuse ❶. If a suitable cache is found, it generates response $\mathcal{R}$ after passing sanity checks ❷. If not, an LLM generates $\mathcal{R}$ ❸. In this case, we store $(\mathcal{P}, \mathcal{R})$ in the cluster database ①. Once enough example pairs accumulate in a cluster, `CodeGenLLM` attempts to generate a program ② and store it in the cache store ③ after validation. Figure 2b shows the prompt for `CodeGenLLM` and the generated program.

share large common templates (*system messages*), which can dominate the embeddings, making semantically unrelated prompts appear similar, considering both prompt and response similarities ensures more accurate clustering. `GenCache` performs online clustering without limiting the number of clusters. Each cluster is maintained as a key-value store, where the key is the cluster ID and the value contains the prompts, their LLM-generated responses, and corresponding embeddings.

**Embedding Generation.** Each prompt $\mathcal{P}$ is converted into an $n$-dimensional embedding $e_p$ using `SentenceTransformer` [37]. However, responses $\mathcal{R}$ are typically JSON-structured rather than plaintext in agentic settings, containing multiple key-value pairs that encode details about the "action" to be performed. To better capture this structure, we create an embedding array $[e_r]$ by computing $n$-dimensional embeddings for each value of the key-value pair of $\mathcal{R}$. If no existing clusters are found, a new cluster $\mathcal{C}_0$ is initialized with $(\mathcal{P}, \mathcal{R})$.

**Similarity Computation.** We compute two similarity scores between a new prompt $\mathcal{P}$ and each cluster $\mathcal{C}_i$, one from prompt embeddings and another from response embeddings. The prompt similarity $s^p$ is the cosine similarity between $e_p$ and the cluster prompt centroid $c_i^p$ (the mean of all $e_p$ embeddings in $\mathcal{C}_i$). The response similarity $s^r$ is the average cosine similarity between $[e_r]_j$ and the corresponding cluster response centroid $[c_i^r]_j$ across all indices $j$. We also ensure that the number of key-value pairs in $\mathcal{R}$ matches that of the cluster response centroid.

**Cluster Assignment.** To assign the input prompt $\mathcal{P}$ to a cluster $\mathcal{C}$, both similarity scores $s^p$ and $s^r$ must exceed their respective thresholds. Let $\mathcal{C}^p$ denote clusters with $s^p > T^p$, and $\mathcal{C}^r$ denote clusters with $s^r > T^r$. Their intersection, $\mathcal{C}^{int} = \mathcal{C}^p \cap \mathcal{C}^r$, represents clusters that meet both conditions. Among these, the final cluster $\mathcal{C}$ is selected as the one maximizing the combined similarity, i.e., $\mathcal{C} = \operatorname{argmax}_{\mathcal{C}^{int}}(s^p + s^r)$.

### 3.3 Program Generation for Cache

For each cluster $\mathcal{C}$, we use an LLM (`CodeGenLLM`) to generate a Python program to be stored as cache. Similar ideas have been explored in prior work [16, 19], though primarily for mathematical reasoning. The generated program takes a prompt $\mathcal{P}$ as input and produces a response $\mathcal{R}$ in the expected key-value format. To generate the program, we provide `CodeGenLLM` with sufficient prompt-response pairs from $\mathcal{C}$ as in-context exemplars [49] (or, exemplars) and instruct it to infer the underlying pattern that maps these prompts to responses. The underlying pattern must identify the variations, which are often keywords from the prompts themselves (e.g., in Figure 1), in the structurally similar prompts and use these to generate the response. Since structurally similar prompts have a specific sentence structure, a regular expression can typically be used to extract those keywords. Thus, when a consistent pattern is detected, `CodeGenLLM` generates a program that extracts relevant keywords using a regex search and constructs the final response accordingly. Returning to our running example Figure 1, `CodeGenLLM` encodes the consistent pattern with a regex `r'buy|purchase|get (item_name) from Amazon'`, constructs the response template as `actions:(1) go to Amazon.com, (2) Search("match.groups(0)")`,

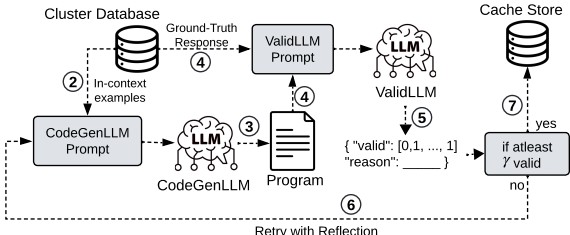

Figure 3: Program validation process before storing it as cache. Step numbering remains consistent with Figure 2a, and we only show from ② onwards here. After program generation ③, ValidLLM validates that the program-generated responses match expected outputs by using in-context examples in the prompt ④. It produces a boolean-array for each example ⑤. If less than $\gamma$ responses match, the system retries cache generation using a reflection-based prompt ⑥. Otherwise, it stores the validated program as cache ⑦.

and (3) press enter', and writes these in a program which is eventually stored as cache. Given user instructions like *"buy {item} from Amazon"*, the program constructs the response by replacing the item name in match.groups(0).

CodeGenLLM's prompt (Figure 2b) includes both the system messages and the user instruction for each in-context example. Including the system message allows the LLM to infer how responses were derived from input prompts. Multiple examples guide CodeGenLLM to identify a consistent pattern, synthesize a program with appropriate regex searches, and capture the necessary variations. The prompt also contains guidelines so that the synthesized program for appropriate error handling, output formatting, and a default {null} response when regex matching fails. This ensure reliable and well-formed outputs, preventing potential disruptions during agent execution. The full prompt used for CodeGenLLM is included in Appendix. GenCache does not impose strict limits on $\mathcal{P}$ or $\mathcal{R}$ length. While short $\mathcal{P}$ and $\mathcal{R}$ often exhibit the two properties discussed in §1, GenCache applies equally well to longer $\mathcal{P}$ and $\mathcal{R}$ generated by automated systems (e.g., SRE agents), as our evaluation shows.

A minimum number of exemplars, $\nu$, is required before invoking CodeGenLLM. If clusters contain more than $\nu$ exemplars, all are used in the CodeGenLLM's prompt. However, each cluster is limited to at most $3\nu$ exemplars (§3.5), and hence the prompt does not grow indefinitely. Larger $\nu$ enables CodeGenLLM to easily identify the patterns, but may introduce more noise from outliers in the cluster. We evaluated varying $\nu$ to study this trade-off. Once a cluster contains at least $\nu$ exemplars, program generation is invoked in the background without delaying the client response.

### 3.4 Program Validation

Before storing a generated program as cache, it is validated for correctness, specifically, whether it can process the input prompt, extract relevant keywords, and generate a response with appropriate variations in the required format. This validation step ensures that only reliable programs or those with proper exception handling are cached.

We use a separate LLM, ValidLLM, to validate each generated program. Using the same exemplars employed during program generation (§3.3), the program is executed locally via a Python interpreter to produce *program-generated responses*. ValidLLM assesses whether each response matches its corresponding exemplar response, both in content and format. For textual components, it checks semantic equivalence and the presence of key information instead of strict matching (e.g., *"The user wants to purchase Item X"* matches with *"user wants to buy item X"*). ValidLLM returns a boolean (0 or 1) indicating correctness for each exemplar, along with a justification. This process verifies that the cached program produces correct program-generated responses for the exemplars.

If fewer than a threshold $\gamma$ of the program-generated responses match their exemplars, we retry program generation using *reflection*, incorporating ValidLLM's justification feedback into the CodeGenLLM prompt. If at least $\gamma$ responses match, the program is accepted and stored as a cache. Figure 3 illustrates this workflow. GenCache limits the number of cache generation retries for each cluster to $\rho$ to control the cost. The complete prompt for ValidLLM is included in the appendix.

The generated program may overfit to specific exemplars (e.g., contain hardcoded keywords). To address this, we (i) prompt CodeGenLLM to include appropriate error handling blocks (§3.3), (ii) verify that the input prompt conforms to the general prompt structure within the cluster using a stored regex before retrieving the cache (§3.1). On failure, GenCache defaults to using LLM for response.

### 3.5 Cache Management

Each cached program is typically under ∼5KB, so the total cache footprint is capped at a few tens of MBs. When the cache is full, the eviction policy discards the least-recently used cache entry while retaining its associated cluster. Clusters are stored in a database allocated a few hundred MBs, though usage remained below 20 MB in all experiments. To prevent unbounded cluster growth, `GenCache` limits each cluster to $3\nu$ prompt–response pairs. Once full, no new entries are added. We also avoid storing the prompts on cache hit, further reducing database size as hit rates improve. A cached program may still fail for a new request. In such cases, `GenCache` may regenerate the program for the corresponding cluster, but the total regeneration retries are capped at $\rho$.

## 4 Experiments

**Setup:** We implement `GenCache` as a library exposing an API interface through which clients issue input prompts. It returns either a cache-generated response or, on a cache miss, invokes an LLM to generate the response. We evaluate `GenCache` in two scenarios: (1) standalone user prompts (§4.1, §4.2) using synthetic data, and (2) integrate with two AI agents (§4.3). While `GenCache` is compatible with agents from any framework, we evaluate with simple prompt-based agents and leave the incorporation with a more performant agent for future work. The first agent is an open-source web navigation agent Laser [31, 8], built on the OpenAI Chat Completion [1] framework. It plans and executes webpage actions like "Search" and "Buy" to fulfil user instructions. The second agent, FLASH [58], is a cloud-operations agent built using LangChain [6] that diagnose recurring system faults. Given an incident description, it retrieves relevant troubleshooting documents via RAG [28] and follows diagnostic steps until the root cause is identified (mitigation is disabled). Both agents use ReAct-style prompting [54]. We use GPT-4o for `CodeGenLLM` and `ValidLLM`, with default parameters $\rho = 30$, $\gamma = 50\%$, $T^p = 0.8$, and $T^r = 0.75$, unless stated otherwise. Task-specific adjustments are made to `CodeGenLLM` and `ValidLLM` prompts as needed. On cache misses, `GenCache` use GPT-4 to generate responses for FLASH, and GPT-4o for all other experiments. We run our experiments on cloud servers with 8-core Intel Xeon CPU and 64 GB memory.

**Datasets:** For experiments with Laser [8], we use the WebShop dataset [53], a simulated e-commerce environment featuring 12000+ crowd-sourced user instructions. Experiments with FLASH use incident diagnosis data from Microsoft, collected over 2 months, comprising 298 incidents across five troubleshooting scenarios. The top-1 troubleshooting scenario accounts for 69% of incidents, and the top-2 for 93%, indicating high recurring tasks that make caching particularly effective instead of repeated LLM calls. For standalone prompt evaluations, since `GenCache` targets repetitive tasks with structurally similar prompts, general language understanding benchmarks [44, 51, 60, 56] that evaluate an agent's comprehensiveness on diverse tasks are not suitable. Instead, we reuse WebShop instructions, augmenting each with a maximum price, resulting in a consistent structure: a purchase verb (e.g., "I want to buy"), followed by an item description with attributes and a price limit. While item details and prices vary across prompts, the overall structure remains stable with only minor phrasing differences, making this dataset well-suited for evaluating `GenCache`. To evaluate scenario (1), we use WebShop's user instructions as seed data and prompt GPT-4o to generate two synthetic prompt sets with the below characteristics. These prompts are then issued to `GenCache`'s API interface with the expected response of receiving just the (item name, price limit) tuple.

1. *Param-Only*–only the parameters vary while phrasing remains identical. We use GPT-4o to extract the item description (LLM is allowed to rephrase this) and price. We then reformat the instruction using the template *"I want to buy* {item}, *under the price range of* {price}*

2. *Param-w-Synonym*–both parameters and the verb phrasing vary but the `verb-item-price` semantics is maintained. We use GPT-4o to rephrase the instruction by adding some optional words (like "please") at the beginning for a few prompts or splitting the sentence into two. This alters the structure slightly without chaning the semantics.

**Baselines:** We compare `GenCache` to `ExactCache` and *GPTCache* [14, 4], a widely popular representative among semantic caching techniques. For `ExactCache`, we leverage hash-based indexing of prompts. GPTCache, however, uses embeddings similarity for prompt matching. We use the Sentence Bert embedding `all-MiniLM-L6-v2` [37] for computing prompt embeddings and the FAISS [25] indexing strategy for similarity search with a similarity threshold of 0.95. We do not compare with prompt-prefix matching techniques [13, 22] as they focus on low-level decoding operations and are orthogonal to `GenCache`. Our baseline choices are well-detailed in the supplementary material.

| Method | Param-Only | | | Param-w-Synonym | | |
|---|---|---|---|---|---|---|
| | Hit % | +ve Hit | -ve Hit | Hit % | +ve Hit | -ve Hit |
| ExactCache | 0 ($\pm$ 0.0) | N/A | N/A | 0 ($\pm$ 0.0) | N/A | N/A |
| GPTCache [14] | 90.92 ($\pm$ 0.02) | 0 ($\pm$ 0.0) | 100 ($\pm$ 0.0) | 88.71 ($\pm$ 0.01) | 0 ($\pm$ 0.0) | 100 ($\pm$ 0.0) |
| GenCache | **97.81 ($\pm$ 0.96)** | 98.03 ($\pm$ 0.05) | 1.97 ($\pm$ 0.05) | **83.66 ($\pm$ 6.37)** | 92.16 ($\pm$ 0.12) | 7.84 ($\pm$ 0.12) |
| GenCache-feedback | 82.35 ($\pm$ 1.57) | **99.63 ($\pm$ 0.02)** | **0.37 ($\pm$ 0.02)** | 68.32 ($\pm$ 4.85) | **95.58 ($\pm$ 0.08)** | **4.4 ($\pm$ 0.08)** |

Table 1: Baseline comparison of Hit Rate and its correctness with different prompt datasets

We evaluate `GenCache` in the following categories:

1. What is the cache hit rate, and how many times does the hit result in an incorrect response?
2. What is the overall cost in terms of the number of LLM calls and the token sizes to create a reusable cache? Does our cache generation cost exceed the savings we get?
3. How well does `GenCache` perform within repeatable agentic workflows?

## 4.1 Hit-Rate Measurements

We measure the cache hit rate of each method over 10,000 synthetic input prompts, and report the results in Table 1. Each experiment starts with an empty Cache Store and Cluster Database. The first $\nu$ input prompts (default $\nu = 4$) are therefore used to populate the clusters before caching becomes possible. In addition to cache hit rate, we evaluated the response correctness generated by `GenCache` using `GPT-4.1` with 5% responses validated through human feedback. `GPT-4.1` checks whether the cached item's name and price semantically match the ground truth and whether the item remains searchable in an e-commerce context. This indirectly measures the reliability of `ValidLLM`'s semantic comparison. Each cache hit is classified as either a **+ve Hit** (semantically correct item name, attributes, and price) or a **-ve Hit** (incorrect or unsearchable entry). We also extend `GenCache` to reduce negative hits by incorporating feedback from `GPT-4.1` that acts as an oracle semantic correctness verifier. We name this method `GenCache`-feedback. When a cache-generated response is identified as a negative hit, `GenCache` deletes the corresponding cached program, but retains the associated cluster so that it can retry generating a new program.

We observe that `GenCache` achieves over 97% cache hit rate with $\sim$2% negative hits for *Param-Only* dataset. Since only the parameters (i.e. item name and price) vary, `CodeGenLLM` effectively learns reusable patterns that can be cached as programs. The few negative hits arise mainly from long or complex item descriptions that lead to suboptimal e-Commerce search results, as flagged by `GPT-4.1` and human feedback. For *Param-w-Synonym* where verbs phrases may deviate, the cache hit rate drops to 84%, with a higher false positive rate. Beyond long item descriptions, false positives also arise due to cases where the entire user instruction is broken into multiple sentences. For example, a cached pattern like `r'buy|need|purchase|get (item name)'`, misfires on the user instruction *"want a wireless headphone. need it in black"*, returning *"it in black"* as the item. Incorporating feedback in `GenCache`-feedback slightly reduces the overall hit rate due to cache deletions but also decreases negative hits. This shows that if a client (or an agent) can provide a feedback that the response from cache use resulted in an error, `GenCache` can adapt and modify the cache.

`ExactCache` exhibits no cache hits, while all GPTCache's cache hits are negative. Since there are no prompt repetitions, no responses should be repeated as well. However, GPTCache returns a cached response corresponding to a prompt that shows high similarity to the new request (example in §1).

## 4.2 Cache Generation Cost vs Savings

`GenCache` incurs an initial cost for using `CodeGenLLM` and `ValidLLM` to generate and validate programs before caching. For caching to be effective, this cost must remain lower than the savings from subsequent cache hits. To evaluate this trade-off, we compute the ratio of LLM calls used for cache creation (cost) to the number of cache hits (savings, i.e., avoided LLM calls). We run `GenCache` with the default $\nu = 4$ on 5,000 input prompts for each dataset type and plot this ratio over time in Figure 4. Initially, the ratio exceeds 1 as `GenCache` incurs setup costs to build caches. Once the number of prompts surpasses $\nu$, cached programs are reused, sharply increasing cache hits and reducing the ratio. The *Param-Only* dataset achieves a lower ratio due to higher cache reuse, while *Param-w-Synonym* shows a higher ratio due to more variable phrasing.

While Figure 4 shows cost for a fixed $\nu$, varying $\nu$ changes the number of in-context exemplars for `CodeGenLLM`, affecting the LLM calls needed to generate reliable, reusable caches. As shown in Table 2 for *Param-w-Synonym* prompts across five runs, increasing $\nu$ consistently reduces LLM calls.

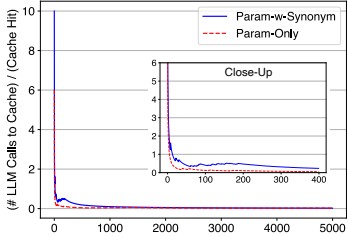
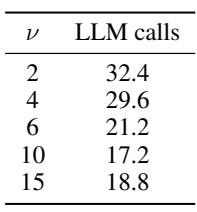
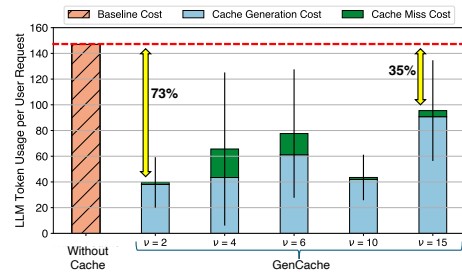

| $\nu$ | LLM calls |
|---|---|
| 2 | 32.4 |
| 4 | 29.6 |
| 6 | 21.2 |
| 10 | 17.2 |
| 15 | 18.8 |

Figure 4: Ratio of no. of LLM calls used for creating cache to number of cache hits plotted against incoming prompts in time

Table 2: Total no. of LLM calls to create reusable caches

Figure 5: LLM token usage for caching compared against the baseline of 'Without Cache'. With no prompt repetitions, `ExactCache` incurs same cost as the baseline

| **Cache Hit Workflow** | Time (s) |
|---|---|
| Cache lookup (includes identifying similar cluster) | 0.112 |
| Cache retrieval & local program execution (reusing the cache) | 0.064 |
| **Cache Miss Workflow** | Time (s) |
| Cache lookup (to detect miss) | 0.056 |
| LLM call (to generate a new response) | 3.520 |
| Database insertion (to store the prompt-response pair) | 0.075 |

Table 3: Cache Miss and Hit Workflow Times

More exemplars help `CodeGenLLM` detect recurring patterns and produce reusable programs. Contrary to expectations of diminishing returns, the structural similarity and consistency of prompts reduce noise, so additional exemplars continue to improve pattern reliability. This trend likely generalizes across domains, though longer prompts slightly increase program generation cost.

In addition to reducing the number of LLM calls, `GenCache` must also minimize token usage, as LLM pricing typically depends on input and output tokens [9]. To evaluate token usage cost, we vary $\nu$ and compute the average tokens (input + output) used per user request during cache creation across five runs (each with 2,000 prompts), comparing it against the cost from cache misses (Figure 5). As a baseline, we plot the token usage without caching. Across all $\nu$, caching yields at least 35% token savings per request, reaching up to 73% for $\nu = 2$. As $\nu$ increases, cache construction cost rises due to longer in-context prompts for `CodeGenLLM`, but plateaus quickly as cache hits become more frequent than cache creations. The cost dip at $\nu = 10$ is because a highly reused cache was stored with very few retries. Figure 5 also shows large standard deviations in cache construction and cache miss cost, mainly due to `GPT-4o`'s variability across runs, which caused multiple retries in some cases. Improved prompt engineering like optimized prompting, ewer repeated guardrails, etc. or using a stronger model like `GPT-4` can further reduce cache construction cost.

**Breakdown of Computational Overheads:** A call to `GenCache`'s API interface follows different steps depending on whether a cache hit or a miss occurs. Each call begins with a *cache lookup* to identify whether a cache is present or not. The lookup consists of:

1. *Embedding generation*: converting the user prompt into an n-dimensional vector ($\sim 0.042$s)

2. *Cluster similarity search*: matching the embedding to the nearest cluster ($\sim 0.009$s)

3. *Regex Validation*: regex-based prompt validation before using the cache, and bookkeeping

During a cache hit, all the above three steps for cache lookup are executed, followed by retrieving the cache and running the program locally. The total latency is $\sim 0.176$s (detailed breakdown in Table 3). Embedding generation accounts for 22.16% of the cache hit workflow, while cluster similarity search adds 4.5%. In a cache miss, regex validation is skipped during cache lookup because no cached entry is found. However, the lookup overhead (0.056s) and database insertion of the prompt-response pair ($\sim 0.075$s) add only $\sim 2.1\%$ overhead, while an LLM call issued on a cache miss takes $\sim 3.5$s on average. Overall, clustering and cache reuse incur negligible overhead while providing significant savings over using an LLM call to respond to a user request.

### 4.3 Impact on Agentic Workflows

We evaluate end-to-end performance impact of `GenCache` on two agentic workflows. Since such workflows already leverage `ExactCache` by default, we deploy `GenCache` alongside it to measure

| Agent | Performance Metrics | GenCache+ExactCache | ExactCache |
|---|---|---|---|
| FLASH | No. of API calls | 4725 | 4960 |
| | No. of LLM calls for Cache Creation | 536 | N/A |
| | % Cache Hits (`ExactCache` contribution) | **54.7%** (31.1%) | 34.4% |
| | Execution Time | **39.91s** | 53.45s |
| Laser | No. of API calls | 1315 | 1308 |
| | No. of LLM calls for Cache Creation | 100 | N/A |
| | % Cache Hits (`ExactCache` contribution) | **37.2%** (12.8%) | 5.7% |
| | Execution Time | **5.02s** | 5.16s |

Table 4: Evaluating the benefits of using `GenCache` in regular agentic workflows

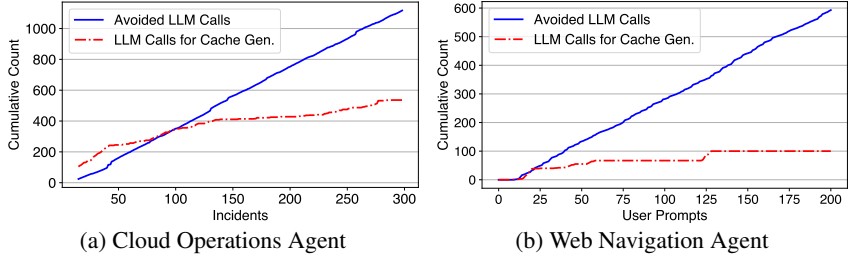

(a) Cloud Operations Agent    (b) Web Navigation Agent

Figure 6: Comparison of LLM calls avoided due to cache hit (savings) with the LLM calls incurred for cache creation (cost). As more inputs are processed, the gap between savings and cost widens.

additional benefits. Table 4 summarizes improvements in cache hit rate and execution time, while Figure 6a compares the associated cost and savings.

**Cloud-Operations Agent:** To diagnose an incident, `FLASH` [58] took 16 LLM calls on average, including identifying the correct documents, generating a coarse and fine-grained plan, and conducting the diagnosis. `GenCache` improved cache hit rate by 23% achieving 54% overall, compared to ∼34% with `ExactCache` alone. Using `GenCache` also reduced the average diagnosis time of the agents by ∼25%. Figure 6a illustrates that as more and more incidents are handled by `FLASH`, the number of LLM calls avoided due to cache hits outweighs the LLM calls incurred to create cache which plateaus. The growing gap demonstrates `GenCache`'s long-term benefits. Out of 536 LLM calls made for cache creation, only 10.4% (56/536) resulted in reusable caches within the workflow; the remaining attempts were flagged as invalid by `ValidLLM`. There were also failure cases; across 298 incidents, `FLASH` failed in only 3 cases (1%) due to mapping to an incorrect troubleshooting document as a result of cache hit, but it was non-detrimental since we focused only on reversible workflows.

**Web-Navigation Agent:** We run Laser [31] on 200 requests from the WebShop dataset [53], which iteratively queries the LLM, first for a *rationale*, then for the corresponding action type. The workflow is typically to search for an item, compare item descriptions to user requirements, and ultimately add it to the cart. Since rationales involve prompt-specific reasoning and are structurally diverse, we disable `GenCache` for those steps. Instead, we enable `GenCache` only when the LLM selects an action type and its parameters. On average, Laser required 12 LLM calls per request, of which ∼50% (1315 total) used `GenCache`'s API interface. Using `GenCache`, we observed 37.2% cache hits (489/1315), with 12.8% of them (63/489) due to exact prompt matching. Similar to `FLASH`, we see that the savings due to cache hit grow with more prompts while LLM calls for cache generation plateaus in Figure 6b. `GenCache` incurred an additional 100 LLM calls for cache creation. Out of these 100 LLM calls, 34% resulted in creation of reusable caches, while 66% failed (number of retries capped at $\rho = 30$) since the exemplars that were clustered together were too different. Failure cases were: (i) 5.5% of prompts failed due to the cached program repeatedly returning the wrong item from the catalog, and (ii) 11% failed due to formatting errors in cached response–issues that can be addressed through improved prompting, output guardrails, or providing feedback to `GenCache`. Long and complex item descriptions, which were flagged as negative hits in Table 1, did not lead to failures here, since the multi-step workflow included intermediate LLM calls for generating rationales, which mitigates the impact of complex item descriptions.

### 4.4 Sensitivity Analysis

**Similarity Threshold for Clustering:** To analyze the effect of cluster similarity thresholds on

cache hit rate, we varied $T^p$ and $T^r$ (see §3.2). Experiments were conducted on the *Param-w-Synonym* dataset with $\nu = 4$ and $\gamma = 50\%$. We observe in Table 5 that increasing $T^p$ and $T^r$ reduced the cache hit rate, as fewer prompts were clustered together, resulting in many small clusters with insufficient exemplars to generate reliable programs. Either new prompts were often assigned to clusters that did not have any associated cached programs, or in cases where the clusters had existing caches, they exhibited overfitting. In contrast, a low threshold grouped diverse prompts into a single cluster, making it harder for `CodeGenLLM` to learn consistent patterns, again reducing cache hits. Overall, the cache hit rate remains high within a moderate range of similarity thresholds as shown in the table.

| $T^p$ | $T^r$ | Hit % |
|---|---|---|
| 0.75 | 0.7 | 79.72 |
| 0.75 | 0.75 | 84.48 |
| 0.8 | 0.75 | 83.66 |
| 0.85 | 0.85 | 60.02 |

Table 5: Hit Rate vs similarity threshold

**Sensitivity to $\gamma$:** We vary $\gamma$, the threshold for the proportion of exemplars that must match the program-generated response, and evaluate on *Param-w-Synonym* dataset using the same setup as in Table 1. We observe in Table 6 that increasing $\gamma$ improves cache hit rate, indicating that at a lower $\gamma$, the generated program was not general enough to cater to all the minor variations in the prompt. As $\gamma$ increased, `CodeGenLLM` created reliable programs by considering more patterns in the regex. At 50% threshold, the generated program may have resulted in more cache misses than at 70% or 90%, but we observed that the positive hit rate remains similar. Thus, even though the recall is lower, the precision remains high. However, with higher $\gamma$, more LLM calls are required early in the run, when the database contains only a few prompts, to create reliable caches. Once sufficient prompts are clustered and cached programs are generated, cache misses become rare.

| $\gamma$ | Hit % | +ve Hit |
|---|---|---|
| 40% | 80.53 | 91.74 |
| 50% | 83.66 | 92.16 |
| 70% | 89.8 | 92.47 |
| 90% | 94.8 | 94.74 |

Table 6: Hit Rate vs $\gamma$ (threshold for matching responses during validation)

## 5   Related Works

**Semantic Caching:** While GPTCache [14] is a popular semantic caching approach, few works have focused on implementing a caching architecture to make semantic caching usable in real-world settings [17, 29, 35, 63]. SCALM [29] identifies requests frequently visited by users and selectively caches those requests, while [63] improves LLM inference by introducing model multiplexing along with semantic caching. However, [63] relies on the existence of some semantic caching oracle that can group prompts without false positives. LangCache [20] innovates on the embedding layer by domain-adaptive fine-tuning of the embedding models. MeanCache [21] introduces a user-centric semantic caching system that preserves user privacy, and hence employs federated learning to build different embedding models locally at each user device. InstCache [64] introduces predictive caching for short user prompts by predicting user instructions using LLMs and pre-populating the cache.

**Prompt Caching:** Reusing attention states using Key-Value (KV) Cache is a popular LLM inference optimization for a single prompt during autoregressive token generation [36]. Prompt caching [22, 52, 13, 61, 55, 30] extends this idea to multiple prompts, where KV caches across multiple prompts are reused based on prompt prefix matching. Prompt Cache [22] designs an explicit structure for writing prompts to enable seamless detection of prompt prefixes. SGLang [61] and ChunkAttention [55] build efficient data structures for KV cache reuse, while works like Cache-Craft [13] and Cache-Blend [52] implement prompt caching for RAG systems by efficiently reusing and recomputing only the necessary cached chunks. Prompt caching is also used in popular LLM services [10, 3] to reduce costs. However, these works are orthogonal to `GenCache`, and they can be used in parallel to reduce costs when `GenCache` incurs a cache miss.

## 6   Summary and Discussion

We proposed `GenCache`, a novel caching technique for structurally similar prompts that uses LLM to identify a common pattern to generate responses from similar prompts, and caches the patterns as programs after validation. On a cache hit, a stored program is executed to generate variation-aware responses. Future works include supporting structurally diverse prompts and non-reversible agent workflows, a limitation of the current work. Furthermore, modifying AI agents to identify negative hits and relaying back the feedback to the caching layer can help improve cache accuracy over time. Designing structured human-LLM interaction schemas, like in [22] will enable better caching.

# 7 Acknowledgements

This work was supported in part by the Illinois Distinguished Fellowship from UIUC. We further appreciate the anonymous reviewers for their valuable and constructive feedback that greatly improved the manuscript.

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

# Appendix

Here, we provide detail about the following topics:

1. Code Link
2. Choice of Baselines
3. Limitations of `GenCache`
4. Prompts for the LLMs used
5. Data for `FLASH`
6. Structural Modification to the Data
7. LLM token usage pattern over time

## Code Link

Code link for `GenCache` is available at https://github.com/sarthak-chakraborty/GenCache

## Choice of Baselines

`GenCache` stores the input prompt along with the generated program in its cache store. The program can then be executed locally via a runtime like Python interpreter to generate the correct response for the input prompt. Thus, the obvious candidate for a baseline is the exact prompt matching (`ExactCache`), which returns the same response verbatim when the input prompt is identical.

Semantic caching is a widely used technique for LLMs where input prompts are matched to stored prompts based on embedding similarity, and a cache hit returns the exact response associated with the matched prompt. There are multiple semantic caching techniques in the literature. We chose GPTCache [14] as the representative approach due to its wide popularity and well-maintained library (over 7500 stars on Github [5]). Other approaches, such as Mean Cache and the method proposed by Gill et al.[20], improve semantic caching by customizing the embedding model via fine-tuning on domain- and user-specific data. However, these methods are not open-sourced and, like GPTCache, suffer from a key limitation that in datasets without prompt repetition, any cache hit results in a negative hit. InstCache [64] is a recent predictive caching technique that predicts tokens likely to appear in an input prompt and precomputes responses for different token combinations using an LLM. Since InstCache does not have an available open-source code, reproducing its functionality is infeasible without knowledge of the exact prompting strategy used.

Zhu et.al. [63] introduce caching with model multiplexing to improve LLM inference. However, their caching technique is primitive, they check whether the request id is present in the cache or not. If so, it uses the same response verbatim, otherwise, it uses LLM to generate the response and chooses to add the request to the cache based on a novel cache replacement policy. Since `GenCache`'s main technique is generating an appropriate reusable cache, rather than a cache replacement policy, [63] is not an ideal baseline.

Prompt prefix matching is a common caching technique where, if a new prompt shares a prefix with a stored prompt, the key-value (KV) caches of the stored prompt are reused to accelerate inference. Many works leverage this idea [22, 10, 26, 18, 55]. Other works like Cache Blend [52] and CacheCraft [13] extend this idea to adapt within the RAG framework, while also selectively recomputing attention states when prompt prefixes differ slightly. However, these techniques operate at the level of KV cache reuse and represent low-level decoding optimizations, making them orthogonal to our approach. While such methods reduce inference latency, `GenCache` focuses on reducing the frequency of LLM calls altogether. That said, prompt prefix matching can be integrated into `GenCache` as an optimization during cache misses, when LLM invocations are required anyway.

## Limitations

We scope `GenCache`'s applicability to AI agents that employ structurally similar prompts for repetitive tasks, such as fault diagnosis, web navigation, and computer-using agents. We find that `GenCache` is particularly effective for agents following ReAct-style prompting [54], especially when LLM responses correspond to actions rather than rationales, as these exhibit consistent structural patterns across invocations. `GenCache` also performs well when variable user instructions are generated automatically (e.g., by alert monitors in SRE agents), since such prompts naturally maintain high structural similarity.

Since `GenCache` relies on LLMs to identify consistent patterns and store them as reusable caches, `GenCache` may identify an incorrect cache for a few prompts and consequently generate erroneous responses. Therefore, we recommend using `GenCache` with strict guardrails and mechanisms to detect such errors. When an agent detects an incorrect cache-generated response through downstream validation, it can provide feedback to `GenCache`, which then deletes the corresponding cache entry (`GenCache`-feedback). Hence, agent reliability and the ability to identify or roll back incorrect executions are critical for ensuring `GenCache`'s robustness in practice.

Although `CodeGenLLM` can identify branching patterns and generate programs with `if-else` clauses when provided with sufficient exemplars, `GenCache` may fail when the number of branches is large, as too few exemplars exist to capture decisions for all branches. Hence, an adaptive strategy is needed to modify the cached program dynamically whenever a new branch is detected, either through agent feedback upon failure or when a cache remains unused.

Future works include expanding `GenCache` to more general prompting structure and agent designs. Prompting strategy for `CodeGenLLM` and `ValidLLM` could be improved to reduce the number of LLM token usage for generating cache. Some common strategies that can be used to improve the prompting is Reflexion [42], Chain-of-Thought [48], Plan-and-Solve [46], self-critic [23], etc.

**Prompts for the LLMs used**

Note that the prompts are not optimized, and there will be multiple repetitions in the instructions provided. A more optimized prompt conveying the same message will improve the LLM token usage for `CodeGenLLM` and `ValidLLM`.

---

**`CodeGenLLM` Prompt (for Web-Navigation Agent and with WebShop dataset; clients use OpenAI Chat Completion API to interact with the LLM)**

You are an expert who can analyze a few example prompts and their corresponding responses and find a common intent from which the responses were generated by the prompts. Once you find the common intent, your task is to generate a Python program for it.

The example prompts will have a 'system' role, a 'user' role and might have a 'function' description. Your task is to analyze the 'text' part within the 'content' subfield of the 'user' role to find a common pattern across the examples. The part that is to be analyzed will contain some static phrases and some variable phrases. You need to understand how these parts relate to the response. If the responses across all examples are similar (e.g., searching a product), the common pattern must be able to identify how to structure the response based on the static and the variable parts of the phrase.

Your job is to:
1. Identify a consistent pattern across examples
2. Generate a Python program that extracts key variables (e.g., item name, attributes, price) using regex and constructs the response accordingly.
3. Ensure the program works for arbitrary prompts that follow the same overall structure, even with minor variations or synonyms.

Here are some guidelines for pattern extraction:
1. Write a regular expression that can extract the variable parts of the phrase from the user instruction
2. Find reusable structures in the prompts, e.g., "buy {item} from Amazon". Identify and divide the human prompt into verb, item and price phrases.
3. For each part of the phrase, write regex containing synonyms using the examples provided (e.g., if the prompt says "find me", synonyms are "i want to buy" or "i am looking for"). Use up to a MAXIMUM of 5 synonyms per phrase (no more), or the code may raise EOL errors.
4. Use the synonyms to write a regex to identify the item along with its attributes, and the price of the item
5. Generate regex that captures most prompts and is general enough to apply to new prompts, not just the seen ones (e.g. for keyword extraction, identify where the keyword appears in the sentence and extract them from the similar part of the sentence for any arbitrary user

---

instruction, but following the same prompt template). Please go through all the examples provided before coming up with the regex pattern.

6. Use `re.DOTALL` if needed (e.g., multi-line string matching).

Example Input Prompts and their Corresponding Responses
================================================

Guidelines for the code output format and other generic guidelines:

1. Analyze only the 'text' portion in the 'content' subfield of the 'user' role.

2. The code must produce an output in the exact format shown in the example responses. For responses in the format of a dictionary, there should be no additional key-value pairs, otherwise, there will be errors in the downstream task that uses this output.

3. Put escape characters for single and double quotes wherever necessary to avoid syntax errors.

4. Replace `\\n` with `\n` in the final code to handle newline characters correctly from the command line input of the prompt.

5. Put ample `try/except` blocks to catch errors. The code should not crash for any input prompt. If any error occurs, print 'None' and the error.

6. Every `if/elif` must be followed by an `else` to handle unmatched conditions. If no conditions are satisfied, the `else` block should print 'None' along with the reason.

7. The code should be complete with no EOL or syntax errors.

Guidelines for Code Execution Format:

1. The code generated will be saved as 'runnable_code.py'.

2. It will be run as 'python3 runnable_code.py <input-prompt>'

3. The code must execute without any manual intervention and take the entire prompt (save it in the variable name 'prompt') as command-line input <input-prompt> which includes both the fixed and the variable parts

Final Instructions (strict):

1. Output only the complete Python code.

2. Do not print any explanation, description, or English text apart from the code

3. The output format should exactly match the format of the example responses.

Now Begin!!

---

### `CodeGenLLM` Prompt (for Cloud-Operations Agent; clients use Langchain API to interact with the LLM)

You are an expert who can analyze a few example prompts and their corresponding responses and find a common intent from which the responses were generated by the prompts. Once you find the common intent, your task is to generate a Python program for it.

The example input prompts provided below will have a prompt template, describing what the LLM was asked to do. This is the static part. It will also contain a dictionary of inputs where each key-value pair can be represented as '{abc:def}'. To reconstruct the actual prompt that was used to query the LLM, replace each key ('abc') in the template with its corresponding value ('def'). The 'def' part in the full prompt is the variable part. Your goal is to analyze how this transformed prompt maps to the given output and find a generalizable pattern that applies across all examples. If the responses across all examples are similar (e.g., calling the same API with some parameters), the common pattern must be able to identify how to structure the response based on the static and the variable parts.

Your job is to:

1. Identify a consistent pattern across examples

2. Generate a Python program that extracts key variables using regex and constructs the response accordingly.

3. Ensure the program works for arbitrary prompts that follow the same overall structure.

Here are some guidelines for pattern extraction:
1. The common pattern can be a code to perform a task (extracting a substring from a string) or even a text sentence if all the example responses are sentences with minor changes that do not alter the semantics.
2. For extracting a substring from a string, strip leading/trailing whitespace from the string before matching.
3. Generate regex that captures most prompts and is general enough to apply to new prompts, not just the seen ones (e.g. for keyword extraction, identify where the keyword appears in the sentence and extract them from the similar part of the sentence for any arbitrary user instruction, but following the same prompt template). Please go through all the examples provided before coming up with the regex pattern.
4. Use `re.DOTALL` if needed (e.g., multi-line string matching).

In the examples below, the input dictionary is written in the form:
```
KEY -> ABC
VALUE -> def
```

Example Input Prompts and their Corresponding Responses
================================================

Guidelines for the code output format and other generic guidelines:
1. The code must produce an output in the exact format shown in the example responses without 'Thought'. For responses in the format of a dictionary, there should be no additional key-value pairs, otherwise, there will be errors in the downstream task that uses this output.
2. Do NOT include 'Thought' in the code-generated output, even if it appears in the examples.
3. Put ample `try/except` blocks to catch errors. The code should not crash for any input prompt. If any error occurs, print 'None' and the error.
4. Every `if/elif` must be followed by an `else` to handle unmatched conditions. If no conditions are satisfied, the `else` block should print 'None' along with the reason.
5. The code should be complete with no EOL or syntax errors.

Guidelines for Code Execution Format:
1. The code generated will be saved as 'runnable_code.py'.
2. It will be run as 'python3 runnable_code.py <input-dict>'
3. The code must execute without any manual intervention and take the input dictionary in the form '{abc:def}' passed as a string as command-line input <input-dict>.

Final Instructions (strict):
1. Output only the complete Python code.
2. Do not print any explanation, description, or English text apart from the code
3. The output format should exactly match the format of the example responses.
Now Begin!!

## ValidLLM **Prompt**

You are an expert evaluator tasked with comparing multiple LLM-generated outputs to their corresponding ground-truth answers. Each answer may be a JSON object, a string, or a code snippet. You should validate only the JSON or code portions; ignore any general English descriptions.

Some of the comparisons may be about an API call searching for a product description that users want to buy. In those cases, consider a match valid if the key attributes are preserved, even if phrased differently (e.g., "a blue headphone with active noise cancellation" and "blue headphone, active noise cancellation"). Often, the LLM-generated answer will be more verbose (former in the example) than the ground-truth answer (latter in the example).

Some important validation rules are:
1. If the ground truth response is in the JSON format, all keys must be present in the LLM-generated response as well. Extra keys in the JSON mean the result is invalid.
2. If some values within the JSON contain English sentences, check semantic equivalence between the ground truth and the LLM-generated response, not exact wording (e.g. "The product is available in the store" and "The store has the product available" are semantically equivalent).
3. For verbose (in LLM-generated response) vs. concise (in ground-truth response) sentences when comparing for certain keys in the JSON, ensure keywords from the concise form appear in the verbose one.
4. For short phrases or code blocks in the response (e.g., "Buy Item", "Search"), check for exact matches.
5. If the LLM-generated response contains 'null' for some keys in the response, while the ground-truth response contains 'None', treat them as equivalent.
6. Ignore punctuation or numeric formatting (e.g., 10 and 10.00 are equal) when comparing.
7. Ignore quote style (single vs. double quotes) (e.g., "content" and 'content' are valid).

Expected Output Format:
```
{
    "valid": [0 or 1],
    "reason": "The output is correct/incorrect because ..."
}
```

"valid" is a list of 0s and 1s (length of the list = number of comparisons done), where 1 at i'th position means a correct match for comparison 'i', and 0 means a mismatch.

"reason" should give a single combined explanation for why any outputs were incorrect (e.g., extra keys, wrong structure, mismatched values). Do not provide individual explanations per comparison, nor include the comparison number. If an LLM-generated response for any comparison includes an error/exception (e.g., "None: <error>"), include that reason. The objective of the reason field is to easily identify mistakes and rectify, hence be concise.

Do not output anything except the specified JSON.

Strictly follow the format and rules above. Now validate the given examples.
=====================================

GPT-4.1 **Prompt (used to measure negative hits in §4.1)**

You are an expert at checking the correctness of two phrases. You will be given an instruction, and two phrases extracted from that instruction. One of the phrases is the ground truth answer, while the other is an answer from our algorithm. The instruction is a user request to buy an item within a price limit. Both phrases will contain (item name, price limit) tuples.

Your task is to verify whether the extracted phrase from our algorithm matches the item description in the instruction. If the description contains some important attributes that will be required if it needs to be searched in an e-commerce website, but those attributes are missing in the extracted phrase, then it is not a match.

The ground truth may have a different phrasing of the item description, which is fine, but the phrase from our algorithm should contain all the necessary information about the item.

**Data for FLASH**

For our experiments with the Cloud-Operations Agent FLASH, we show the distribution of the incidents that map to each troubleshooting scenario in Figure 7a. We see that the first troubleshooting scenario (TS-1) covers 69% of the incidents. Hence most incidents' diagnosis strategy remains the same since they map to the same troubleshooting strategy document. We plot the number of repetitions for each incident in Figure 7b. Of the 298 incidents, 253 were unique. We see that 229 incidents never re-occurred, 15 unique incidents recurred twice, while one incident re-occurred 7 times. This shows that using ExactCache cannot produce a high cache hit rate.

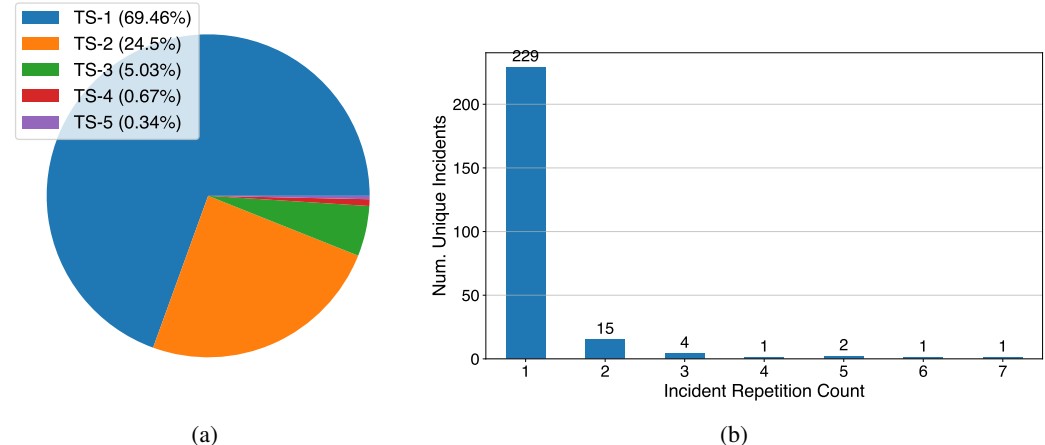

(a)                                                     (b)

Figure 7: (a) Percentage of incidents that map to each troubleshooting scenario, (b) Number of repetitions for each incident

**Structural Modification to the Data**

To complete our experiments in §4.1, we also evaluated on a third dataset with prompt characteristics different from *Param-Only* and *Param-w-Synonym*. We call this variation in the prompt set *structural*, where we expressed each user instruction in 10 different ways with structural variations but semantically identical. For example, "I want to buy Bluetooth headphones, under the price of 150 dollars" is expressed as "For under 150 dollars, I want a Bluetooth headphone".

As shown in Table 7, GenCache experiences a drop in both cache hit rate and precision when there are structural changes in the user instructions, compared to the results in Table 1. The reason for this is that most cached regular expressions fail to generalize when the user instruction structure differs.

|        | Structural |  |  |
| Method | Hit % | +ve Hit | -ve Hit |
| --- | --- | --- | --- |
| ExactCache | 0 ($\pm$ 0.0) | N/A | N/A |
| GPTCache [14] | 96.28 ($\pm$ 0.01) | 20.72 ($\pm$ 0.01) | 79.28 ($\pm$ 0.01) |
| GenCache | 4.23 ($\pm$ 0.21) | 72.4 ($\pm$ 0.12) | 27.6 ($\pm$ 0.12) |

Table 7: Baseline comparison of Hit Rate and its correctness when Prompts had Structural changes

ExactCache has 0% hit rate since no prompts were repeated. Thus, we argue that in domains with structurally diverse prompts, GenCache is less effective (which it is not designed for), and reverting to ExactCache is preferable (hoping that prompts repeat). While GPTCache's negative hit rate continues to be high, it is not 100%, as it occasionally returns correct responses for semantically similar prompts with structural variations. We observe that when GPTCache shows a positive cache hit for one user instruction, it tends to return positive hits for all structural variants of that instruction. However, as the number of user instructions increases, its reliance on approximate nearest neighbor search for semantic similarity often yields incorrect matches, leading to inaccurate cache hits. Since GenCache already experiences a high negative hits, we do not experiment GenCache-feedback on *Structural* prompt set.

**LLM token usage pattern over Time**

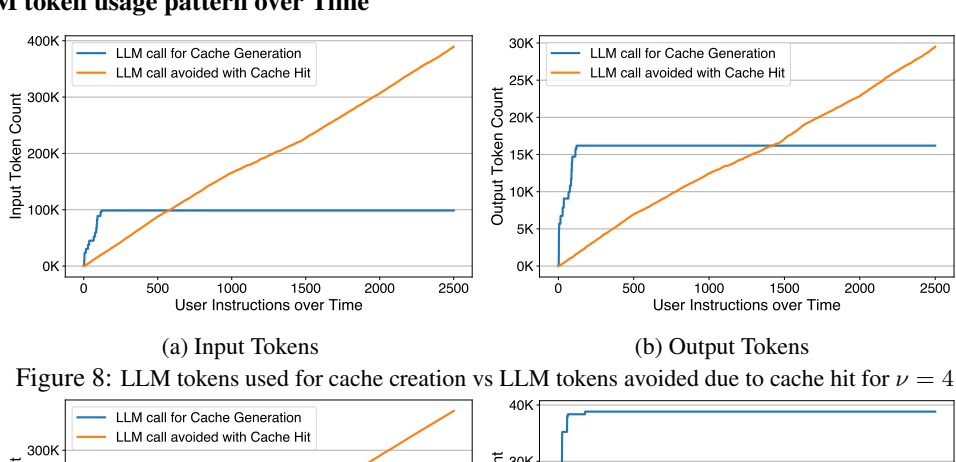

(a) Input Tokens

(b) Output Tokens

Figure 8: LLM tokens used for cache creation vs LLM tokens avoided due to cache hit for $\nu = 4$

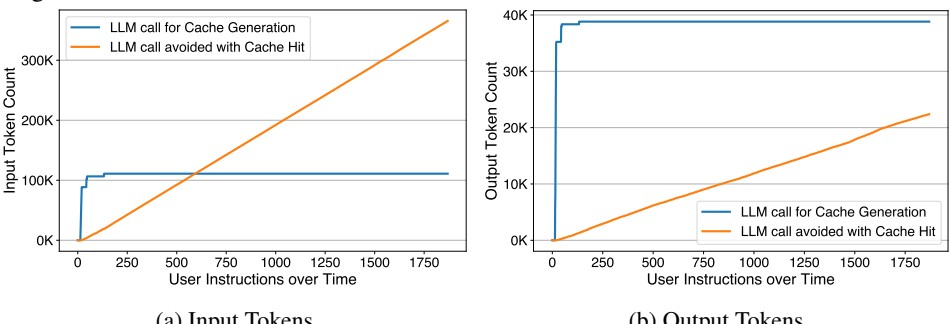

(a) Input Tokens

(b) Output Tokens

Figure 9: LLM tokens used for cache creation vs LLM tokens avoided due to cache hit for $\nu = 15$

In Figure 5, we showed at least 35% token savings per request on using GenCache. We now show how the LLM token usage varies over time. Figure 8 and Figure 9 plots how the number of input and output tokens varies for $\nu = 4$ and $\nu = 15$ respectively. For all the plots, the 'blue' line plots the token usage (input or output) when LLM was called for cache generation, while the 'orange' line plots the tokens (input or output) that were saved as a result of avoiding LLM call due to cache hit. We observe that the LLM token usage during cache generation plateaus after initially being high, while the token savings continue to increase as cache hits become more frequent than cache creations over time. While the benefits due to cache hit in input token usage surpass that of cache generation for both $\nu$, the output tokens used for creating the cache with $\nu = 15$ are still higher than the savings due to cache hit after around 1800 user instructions (even though there is an upward trend).

