# OpenReview forum: "Generative Caching for Structurally Similar Prompts and Responses"
_NeurIPS.cc/2025/Conference — NeurIPS 2025 poster_

### Official Review · Reviewer_6e2p · 2025-06-15

**Clarity:** 3
**Significance:** 3
**Originality:** 3
**Rating:** 4
**Confidence:** 4

**Summary:**

This paper introduces GenCache, a method for leveraging previous LLM responses to handle partially matching prompts in LLM-based web agents that perform repetitive tasks. In a preprocessing stage, GenCache performs clustering of prompts in a dataset and generates a regular expression for detecting and preprocessing prompts in each cluster, and it generates code for common operations for the prompts in each cluster. During inference, it retrieves cached code and adapts it by appropriately substituting variables. Unlike existing caching methods that reuse cached responses without modification, GenCache caches code templates and dynamically adapts them to new inputs, although its current design is tightly coupled to a specific task, which may limit its applicability to broader use cases. Experimental results on the WebShop dataset show a high cache hit rate exceeding 80% and a low negative hit rate below 10%.

**Questions:**

Please let me know if there are any inaccuracies or misunderstandings in the points raised under Weaknesses.

**Ethical Concerns:**

["NO or VERY MINOR ethics concerns only"]

**Final Justification:**

The authors' response confirms that my initial understanding of the paper was largely accurate. Accordingly, **I have increased my confidence in my review and consider the quality of the paper to be above the threshold for acceptance.**

However, I remain concerned that evaluation on a single dataset limits the persuasiveness of the empirical evidence. Therefore, I will maintain my current score and consider "4: Borderline accept" to be an appropriate assessment.

**Limitations:**

Yes.

**Quality:**

3

**Strengths And Weaknesses:**

The main limitation of this paper lies in its narrow focus on the web agent task, and I recommend revising the title to better reflect this scope and avoid misleading readers. Nevertheless, the paper makes a meaningful contribution toward addressing the core challenge of leveraging prior LLM responses for similar prompts and proposes a practical and effective solution. Overall, I consider this work to be above the acceptance threshold.

### Strengths

* This paper targets a common and fundamental challenge of leveraging previous LLM responses for similar or partially matching prompts.
* The proposed method offers a practical and effective approach to leveraging partial matches in LLM caching. Although it is tailored to a specific use case involving web agents performing repetitive tasks, the core idea has the potential to generalize to a broader range of applications.

### Weaknesses

* The method is designed for a specific application

The title suggests that this paper introduces a generalizable method. However, the work specifically targets LLM-based web agents performing repetitive tasks. I recommend either revising the title to more accurately reflect this narrow scope or extending the method to support a broader range of applications.

The proposed method relies on regular expressions to detect and preprocess prompts that match cached patterns, which restricts its applicability to tasks with limited prompt diversity. This constraint significantly limits the generalizability of the approach and is the primary reason I did not assign a higher score.

Although the current method is reasonable as an initial step, I believe the proposed framework has potential for greater robustness. For example, incorporating a preprocessing of prompts or a post-hoc revision step to modify the code using small LLMs could help generalize the method while still maintaining low computational cost.

* Limited evaluation domains

The proposed method is evaluated solely on a single dataset, WebShop, which appears to be particularly well-suited to the strengths of the approach. I recommend conducting evaluations on additional datasets that involve more diverse operations in order to better understand the limitations and generalizability of the method.

---

> ### Author Rebuttal · Authors · 2025-07-31
>
> Thank you for your thoughtful suggestions and insightful reviews. Here are our responses to the weaknesses pointed out.
>
> 1. **The proposed method relies on regular expressions to detect and preprocess prompts that match cached patterns, which restricts its applicability to tasks with limited prompt diversity. This constraint significantly limits the generalizability of the approach**
>
> We agree with the reviewer that GenCache is effective when prompts exhibit three key characteristics: (1) **Structural Regularity:** prompts follow a consistent format, e.g., the item to purchase always appear between fixed phrases like “buy” and “under the price of X dollars.” (2) **Controlled Variability:** while the structure stays the same, elements like item names, fault regions in system alerts, or optional words like “please” can vary. (3) **Predictable Response Patterns:** the responses generated follow a consistent format, even if they are not identical. GenCache is designed to accelerate agents that handle repetitive tasks by leveraging the structure and consistency often found in their prompts and responses.
>
> Currently, most humans interact with agents through a chatbot interface where prompts may have high diversity. While GenCache may not be suitable for free-form chatbot interactions which shows high prompt diversity, we envision that LLM will be integrated into automated workflows where tasks will be repetitive. GenCache is *highly effective* for such use cases. As more agents automate repetitive tasks, such as API orchestration and system diagnostics, GenCache’s applicability will exapnd.
>
> 2. **I recommend either revising the title to more accurately reflect this narrow scope or extending the method to support a broader range of applications.**
>
> Thank you for this thoughtful observation and we will consider revising the title. We agree that the current title may imply a broader applicability than what is demonstrated in the paper. While GenCache is a powerful technique, it is not a universal caching solution for all prompt-response scenarios. Its effectiveness relies on the prompt-response characteristics as described in the answer above. These allow GenCache to generate response suitable for the prompt by reusing cached programs, making it well-suited for repetitive agent workflows.
>
> To better reflect this scope, we will consider narrowing the title accordingly. Additionally, we will include a “Future Work” section outlining how GenCache could be extended or used with other caching solutions to support more general applications.
>
>
> 3. **Incorporating a preprocessing of prompts or a post-hoc revision step to modify the code using small LLMs could help generalize the method while still maintaining low computational cost.**
>
> We agree that incorporating prompt preprocessing or post-hoc code modifications could help extend GenCache to support a broader range of prompt variations. This could enable GenCache to operate effectively even in scenarios with higher prompt diversity. However, identifying when users employ different templates to express semantically similar prompts is challenging, which itself *may* require an LLM call that we wanted to avoid in the first place. This variability can have two key implications:
> (i) Cache creation cost *may* increase, as a wider range of prompt structures makes it harder for CodeGenLLM to detect consistent patterns;
> (ii) Positive hit rate *may* decrease, since mismatches between prompt templates and cached entries may lead to incorrect cache selections.
>
> For GenCache to be effective, we want to maintain a high positive hit rate and low cache creation cost, both of which rely on structural regularity and some predictability in prompts. That said, exploring how to balance generalizability with caching accuracy is a promising research direction for future work, and we plan to investigate this further.
>
> 4. **Limited Evaluation Domain**
>
> Thank you for your suggestion. We agree that evaluating GenCache on a wider range of datasets would strengthen its generalizability. In addition to WebShop, we have also experimented with a proprietary dataset on a cloud-operations agent that diagnoses recurring system faults. This dataset aligns well with our use case, as the prompts are generated from an alert monitoring system and follow a static template, making them ideal for GenCache’s strengths in handling repetitive, structured prompts.
>
> We are actively exploring other datasets, where user tasks are repetitive and prompts follow a consistent format with minor variations. While many public datasets are designed to test agent capabilities across diverse and unstructured prompts, making them less suitable for GenCache, we recognize the value in expanding our evaluation. In particular, we plan to explore datasets such as WebArena and GAIA, which may offer a better fit for our methodology.

---

> > ### Comment · Reviewer_6e2p · 2025-08-01
> > **Re: Rebuttal by Authors**
> >
> > Thank you for your detailed rebuttal. Your response confirms that my initial understanding of the paper was largely accurate. Accordingly, **I have increased my confidence in my review and consider the quality of the paper to be above the threshold for acceptance.**
> >
> > I appreciate the additional information regarding the experiments conducted on a proprietary dataset. However, I remain concerned that evaluation on a single dataset limits the persuasiveness of the empirical evidence. Therefore, I will maintain my current score.

---

### Official Review · Reviewer_7qKt · 2025-07-01

**Clarity:** 2
**Significance:** 3
**Originality:** 3
**Rating:** 4
**Confidence:** 4

**Summary:**

This paper introduces GenCache, a novel generative caching mechanism designed to address the limitations of existing prompt caching techniques for structurally similar LLM prompts with critical subtle variations. Recognizing that recurring tasks in workflows, chatbots, and agents often reuse similar prompts but require variation-aware responses, GenCache identifies and caches validated generative patterns underlying these responses as executable programs. Upon a cache hit, it executes the stored program to efficiently synthesize a new, contextually appropriate response for the new prompt, rather than returning a potentially incorrect static cached response. The authors demonstrate that GenCache achieves a high cache hit rate (83%) with minimal negative hits on datasets lacking exact prompt repetition and significantly improves performance in agentic workflows.

**Questions:**

1. The author mentioned that the overall size of the cache is only tens of Mb, which is very exciting, but I noticed that after the cache is built, the clusters do not disappear, and the clusters contain a large number of dialogue inputs and outputs, as well as vectorized representations of these inputs and outputs. Obviously, the size of this information is much larger than the python files in the cache, and the author should add how much space this information will take up;
2. Other computational overhead: The author experimentally proves that GenCache can effectively reduce the computational overhead caused by LLM reasoning. However, GenCache needs to calculate the n-dimensional vector representation of all inputs and calculate its similarity with all clusters. The author should take this part of the computational overhead into consideration;
3. Experimental details: The experimental setting is unclear. The author did not explain how the specific synthetic prompt set in Table 1 was synthesized, what the specific content is, and how the evaluation indicators were calculated. It is recommended to refer to the explanation of the evaluation indicators in GPTCache to further supplement the experimental details.


Format error: 1) The author mentioned " As shown in Figure 2 with Param-w-Synonym prompts " in line 304, but it should refer to Table 2; 2) This seems to be a more serious error. The author mentioned " The prompt used for ValidLLM is included in the supplementary material." in line 206, but I did not find any appendix in the article. The author also lacked an explanation of the experimental details and prompt content of the paper. Did the author forget to attach an appendix to the paper?

**Ethical Concerns:**

["NO or VERY MINOR ethics concerns only"]

**Final Justification:**

Thanks to the author for the response. It addressed the raised concerns. I have revised my score accordingly.

**Limitations:**

yes

**Paper Formatting Concerns:**

I did not find the appendix in the paper, but the author mentioned in the paper that “The prompt used for ValidLLM is included in the supplementary material.” Did the author forget to upload the appendix?

**Quality:**

3

**Strengths And Weaknesses:**

Strengths:
1. The first dynamic LLM cache: Previous studies could only retrieve static LLM replies by retrieving similar caches. GenCache generates Python code through LLM to achieve more flexible cache reading. Even when facing inputs that have never appeared before, it can extract key information through Python code and generate replies.
2. Higher hit rate and efficiency: Experiments on synthetic data and webshop datasets show that GenCache has a higher hit rate and can effectively reduce the overhead of LLM reasoning.

Weakness:
1. As the author said, Python programs usually implement scalable LLM caches through regular expressions, which means that this method is often only suitable for short dialogue outputs with format requirements. However, regular expression-based methods often have little effect on open dialogues and long text tasks, because such dialogues usually do not have a strict format, and it is difficult to extract information from long texts through regular expressions.

---

> ### Author Rebuttal · Authors · 2025-07-31
>
> Thank you for your thoughtful suggestions and insightful reviews. Here are our responses to the questions.
>
> ## Answer to Questions
> 1. **The clusters do not disappear, and the clusters contain a large number of dialogue inputs and outputs, as well as vectorized representations of these inputs and outputs. Obviously, the size of this information is much larger than the python files in the cache, and the author should add how much space this information will take up**
>
> It is true that we retain the clusters, which take up space. These clusters are necessary at inference time for matching a new prompt with the nearest cluster centroid.
>
> Each cached program is typically under 5 KB and often less than 1 KB. In our experiments, GenCache generated an average of three reusable cached programs per experiment, amounting to roughly *15 KB* of cache. In addition, the database size remained under *20 MB* on running GenCache‑feedback on the Param‑w‑Synonym dataset (Table 1).
>
> To ensure cluster sizes remain manageable, we apply two key constraints: (1) We cap each cluster to hold no more than $3\nu$ exemplars, and (2) We avoid storing prompts that result in cache hits, which helps eliminate redundant data. With higher cache hit rates in the Param‑Only dataset, fewer prompts are stored, resulting in even smaller cluster sizes compared to Param‑w‑Synonym.
>
> 2. **However, GenCache needs to calculate the n-dimensional vector representation of all inputs and calculate its similarity with all clusters. The author should take this part of the computational overhead into consideration**
>
> Thank you for the suggestion, we will consider it while reporting results in the final paper. For experiments on the Param‑w‑Synonym dataset (Table 1), here’s the GenCache API call timing breakdown based on cache hits vs. misses:
>
>
> | **Cache Miss Workflow**                                                 | **Time (s)** |
> |---------------------------------------------------------------------------|--------------|
> | Cache lookup (to detect miss)                                         | 0.056        |
> | LLM call (to generate a new response)                           | 3.520        |
> | Database insertion (to store the prompt-response pair)  | 0.075        |
>
> | **Cache Hit Workflow**                                                                  | **Time (s)** |
> |---------------------------------------------------------------------------------------|--------------|
> | Cache lookup (includes identifying similar cluster)                         | 0.112        |
> | Cache retrieval & local program execution (reusing the cache)     | 0.064        |
>
> The Cache lookup itself consists of:
> (1) *Embedding generation*: converting the user prompt into an n‑dimensional vector (avg 0.039 s)
> (2) *Cluster similarity search*: matching this embedding to the nearest cluster (avg 0.008 s)
> (3) Validating the user prompt by a regex check before reusing the cache, and additional bookkeeping accounts for the remainder
>
> The time taken for cache lookup differs between cache-hit and cache-miss workflows. During a hit, all three steps are executed; during a miss, the regex validation step is skipped because no cached entry is found.
>
> Embedding cost accounts for **22.16 %** (embedding generation cost / (cache lookup + cache retrieval and program execution)) of total cache reuse time, while cluster matching adds **4.5 %**.
>
> 3. **The experimental setting is unclear. The author did not explain how the specific synthetic prompt set in Table 1 was synthesized, what the specific content is, and how the evaluation indicators were calculated.**
>
> We are sorry for the confusion regarding the experimental setting. Here are some clarifications (will add in the final version). To generate synthetic prompts in Table 1, we used WebShop’s user instructions as seed data and prompted GPT-4o to create the required dataset. For example, consider the WebShop user instruction: *“Find me a high-speed dual-style package with a 12\" power amplifier car subwoofer, and price lower than 190.00 dollars.”*
>
> * **Param-Only**: We prompted GPT-4o to extract the item description (e.g., *“high-speed dual-style package with a 12" power amplifier car subwoofer”*) and maximum price (e.g., “190 dollars”). The LLM was allowed to rephrase the item description. We then rephrases the instruction using the template: *“I want to buy {item name with attributes}, under the price range of {price}.”* This produces a new instructions such as *“I want to buy a car subwoofer with a high-speed dual-style package and a 12\" power amplifier, under the price range of 190.00 dollars.”* We applied this to all WebShop user instructions..
>
> * **Param-w-Synonym**: We extracted item name, attributes, and price and re-prompted GPT-4o to rephrase the instruction by adding the optinal word “please” at the beginning for 10% of the prompts or splitting the sentence into two for 5% (e.g., “Find me…car subwoofer. I want it under…190.00 dollars.”). This alters the structure slightly without changing its semantics. We didn’t explicitly enforce synonym replacement (e.g., replace 'buy' with 'purchase') since the WebShop instructions already include diverse phrasings (e.g., “buy”, “find me a”, “I’m looking for”, “I need a”).
>
> **Evaluation Indicators:**
> * *Hit %*: a hit occurs when GenCache identifies a cluster with a corresponding cache.
> * *+ve Hit*: the proportion of hits where the item name, attributes, and price are semantically correct.
> * *-ve Hit*: the proportion of hits where the item name is not semantically correct for  search purposes.
>
> Semantic correctness is evaluated using GPT‑4.1 (distinct from GPT‑4o used for cache construction) by comparing the ground truth exemplar response with the program-generated output. We’ve described this in Lines 270–272, but we will make these definitions more explicit in the final version of the paper.
>
> 4. **Format Error**
>
> 1) We apologize for the oversight and we will correct it
> 2) The appendix was added as supplementary material and was not part of the main pdf
>
> ## Answers to Other Concerns
> 1. **This method is often only suitable for short dialogue outputs with format requirements. However, regular expression-based methods often have little effect on open dialogues and long text tasks, because such dialogues usually do not have a strict format, and it is difficult to extract information from long texts through regular expressions.**
>
> We agree with the reviewer that GenCache is effective when prompts exhibit three key characteristics: (1) **Structural Regularity:** prompts follow a consistent format, e.g., the item to purchase always appear between fixed phrases like “buy” and “under the price of X dollars.” (2) **Controlled Variability:** while the structure stays the same, elements like item names, fault regions in system alerts, or optional words like “please” can vary. (3) **Predictable Response Patterns:** the responses generated follow a consistent format, even if they are not identical. GenCache leverages the structure and consistency in the prompts and responses to accelerate agents workflows that handle repetitive tasks.
>
> Importantly, GenCache *does not impose strict limits on prompt or response length*. Human-written, short dialogue prompts often exhibit structural regularity and controlled variability; however, longer prompts generated by automated systems (like troubleshooting agents) can also show similar characteristics.
>
> Although GenCache is less suitable for free‑form chat interactions with varied prompt structures, it is *highly effective* for a growing set of use cases, e.g., in agentic systems dealing with short use instructions and automated workflows. As more agents automate repetitive tasks, such as API orchestration and system diagnostics, GenCache’s applicability will exapnd.

---

> > ### Comment · Reviewer_7qKt · 2025-08-04
> > **Reply to Author**
> >
> > Thanks to the author for the response. It addressed the raised concerns. I have revised my score accordingly.

---

### Official Review · Reviewer_znMf · 2025-07-03

**Clarity:** 2
**Significance:** 2
**Originality:** 3
**Rating:** 4
**Confidence:** 2

**Summary:**

This paper proposes GenCache, a client-side caching technique for LLMs that clusters prompt–response pairs based on both prompt and response embedding similarities. Once a new prompt’s embedding is close enough to a cluster center, GenCache retrieves the small Python program stored for that cluster and executes it to generate a tailored response.

To build these programs, GenCache uses a CodeGenLLM with a handful of in-context prompt–response examples to infer the consistent mapping pattern, then validates the generated code by running it on the same examples and asking a ValidLLM to semantically compare each program output to its ground truth. If at least half of the outputs match, the program is accepted and cached; otherwise, it’s retried or discarded. Each cached program takes a prompt as input and produces the correctly formatted response, enabling high hit rates with minimal LLM calls.

**Questions:**

1. In line 138, the authors write “Since R often contains multiple key-value pairs… Therefore, we create an embedding array [er] by computing n-dimensional embeddings for each value of the key-value pair of R.” But if R is just free-form text (a sentence or paragraph), how should we interpret “key-value pairs”? Are they really embedding each token, or is this only for structured outputs?

2. Do you apply the same multi-embedding approach to prompts, or do prompts always get a single sentencetransformer embedding? What motivated treating prompts and responses differently, especially since prompts can also be quite long?

3. When you claim that “GenCache performs online clustering.” How much CPU/time overhead does this clustering add compared to simply issuing a new LLM call?

4. Will giving more in-context examples (increasing ν) to CodeGenLLM always improve caching performance, or is there a point of diminishing returns?

5. When you say “using the same exemplars from §3.3, we execute the program locally… to obtain program-generated responses,” are those exemplars the ones inside the cluster or the ones provided as in-context examples to CodeGenLLM?

6. You describe validating a program as “error-free,” yet they accept it if only half of the outputs match. Isn’t that just 50% accuracy—why call it error-free?

7. How reliable is ValidLLM’s semantic comparison in practice? Have you done any human evaluation to confirm its judgments?

8. Even after a program is cached, it may still fail on new prompts. I wonder how frequently does that actually happen in their experiments?

**Ethical Concerns:**

["NO or VERY MINOR ethics concerns only"]

**Final Justification:**

Their response mostly addressed my questions. I main my accept ratings.

**Limitations:**

yes

**Quality:**

3

**Strengths And Weaknesses:**

strengths

1. It captures the real-life setting where prompts are structurally similar yet differ in key details; exact-match or vanilla semantic caches fail here, whereas GenCache’s program cache have the potential to returns the correct, variation-aware response.

2. It introduces a “cache-the-program” strategy—store a concise function that transforms the incoming prompt into the answer instead of caching the raw response—keeping the cache lightweight while generalizing across similar prompts.

3. Strong empirical evidence: GenCache achieves a higher hit rate than the baselines and also achieves latency and token savings.

weaknesses

1. Baselines rely on relatively small-context datasets; it is unclear how the regex-driven program will scale to long, multi-paragraph inputs or to irreversible agent actions?

2. CodeGenLLM and ValidLLM both run on GPT-4o/4, which may be costly; demonstrating that a smaller, open model can handle synthesis and validation would strengthen practicality.

3. Accepting a program when only half of the exemplar outputs match (and still called "error-free") seems brittle and risks incorrect cache hits in downstream use.

---

> ### Author Rebuttal · Authors · 2025-07-31
>
> Thank you for your thoughtful suggestions and insightful reviews. Here are our responses.
>
> ## Answer to Questions
> 1. (and Q2) **But if R is just free-form text (a sentence or paragraph), how should we interpret “key-value pairs”? Do you apply the same multi-embedding approach to prompts, or do prompts always get a single sentence transformer embedding?**
>
> We apologize for the confusion, and we will clarify this in the final version. In our case, LLM responses (R) for prompts (P) are structured as **JSON**, not plaintext, because the agents expect a specific format of responses from the LLM. Accordingly, the prompt itself instructs the LLM to produce responses in that exact JSON schema. This is particularly the case in ReAct prompting, where responses are treated as “actions” that the agent must perform. This is why the prompts and the responses are treated differently, where the responses are encoded in a multi-embedding approach, while we use a sentence transformer for prompts.
>
> 3. **How much CPU/time overhead does this clustering add compared to simply issuing a new LLM call?**
>
> A GenCache API call consists of cache lookup (to identify whether a cache is present or not), retrieving the cache, and running the program locally. The **overall time for GenCache API call** if there is a cache hit is **0.176s** (detailed computation breakup is provided for the response to Reviewer 3, Q2).
>
> On the other hand, the **time to issue a new LLM call** takes **~3.5s** on average, which is issued when a cache miss occurs. During cache miss, the cache lookup overhead is 0.056s, and we take an extra **0.075s** to enter the corresponding prompt-response pair into the database, thus incurring an **extra ~2.1% overhead**. Overall, clustering and reusing the cache incurs negligible overhead while providing significant savings over using an LLM call to respond to a user request.
>
> 4. **Will giving more in-context examples (increasing $\nu$) to CodeGenLLM always improve caching performance, or is there a point of diminishing returns?**
>
> As more in-context examples are provided to the CodeGenLLM, it converges more quickly to generating reliable and reusable programs since it can identify recurring patterns via regex and handle a broader set of prompt variations. Intuitively, we might expect diminishing returns as more in-context examples introduces noise making it difficult for CodeGenLLM to identify a consistent pattern. However, since our prompts exhibit characteristics like (i) prompts follow a consistent format, (ii) while the structure remains stable, certain elements like the item name, price, or optional words like “please” can vary, and (iii) the responses also follow a consistent pattern so that the cached program can generate them by replacing keywords. These characteristics make in-context examples less noisy and hence more examples help in identifying reliable patterns. The trade-off is that increasing $\nu$ increases the prompt’s context length, thus increasing the cost of program generation.
>
> 5. **When you say “using the same exemplars from §3.3, we execute the program locally… to obtain program-generated responses,” are those exemplars the ones inside the cluster or the ones provided as in-context examples to CodeGenLLM?**
>
> We use the same examples that were used to create the program. We want to verify whether the exemplars that were used to create the cache are giving correct program-generated responses or not. However, we do not want the program generation to stall for outliers. Hence, we accept a program as cache if atleast a proportion of exemplars match.
>
> 6. **Accepting a program when only half of the exemplar outputs match (and still called "error-free") seems brittle and risks incorrect cache hits in downstream use.**
>
> We run a sensitivity analysis by varying the threshold for the proportion of exemplars that must match the program-generated response. We choose three threshold values--40%, 50% (default), and 70%, and run GenCache on Param-w-Synonym dataset (1000 user prompts) with the same setting as described for Table 1.
>
> | Threshold | Hit % | +ve Hit |
> | :-----: | :----: | :------: |
> |40% | 80.53 | 91.74 |
> |50% | 83.66 | 92.16 |
> |70% | 89.8 | 92.47 |
> |90% | 94.8 | 94.74 |
>
> We see that increasing the threshold indeed increases the cache hit rate. This indicates that with a lower threshold, the CodeGenLLM was able to create a program, but was not general enough to cater to all the minor variations in the prompt, as the reviewer correctly suggested. As the threshold increased, the CodeGenLLM was able to create a reliable program by considering more patterns in the regex and making the program general for more prompts. At 50% threshold, the generated program may have resulted in more cache misses than at 70% or 90%, but the positive hit rate remains similar. Thus, even though the recall is lower, the precision remains high. However, we observed that with higher thresholds, more LLM calls are required early in the run, when the database contains only a few prompts, to create reusable caches. Once sufficient prompts are clustered and cached programs are generated, cache misses become rare.
>
> We speculate that this sensitivity must also vary with $\nu$, since $\nu$ controls the total number of in-context examples used. With more in-context examples, it may be hard to create a reliable cache when the threshold is high.
>
> As per the suggestion, we intend to do a complete sensitivity analysis with more runs in the final version of the paper.
>
> 7. **How reliable is ValidLLM’s semantic comparison in practice? Have you done any human evaluation to confirm its judgments?**
>
> We do not directly measure the validity of ValidLLM’s semantic comparison. But its reliability indirectly reflects the correctness of GenCache, i.e, if ValidLLM determines that a proportion of program-generated responses matches the exemplar, then the CodeGenLLM-generated program can be reused as a cache. We assess the semantic correctness of the responses generated from the stored program.
>
> We use GPT-4.1 (distinct from GPT-4o used for cache construction) to compare the ground truth exemplar response with program-generated response, classifying each cache hit as positive or negative. Additionally, we sample 5% of cache-hit responses for human evaluation. The combined evaluation (GPT-4.1+human) is reported as positive/negative hits in Table 1. However, we have not carried out a large-scale human evaluation to verify the semantic correctness in practice, though we could report the human-evaluated result separately from the GPT-4.1 evaluated result. We will also make the definitions of the evaluation indicators more explicit in the final version of the paper.
>
> 8. **Even after a program is cached, it may still fail on new prompts. I wonder how frequently does that actually happen in their experiments?**
>
> A validated program stored in the cache can still fail on new prompts in two cases. First, regex matching during cache lookup may fail if the prompt structure differs from the one used during cache generation. In this case, we use LLM to generate the response which is then added to the database for program generation in the future. This behavior corresponds to the cache-miss rate in our evaluation (16.3% for Param-w-Synonym with GenCache, as shown in Table 1).
>
> Second, the regex match may succeed, but the cache-generated response may be incorrect; reflected by the negative hit rate in Table 1. This issue can be resolved if the agent provides feedback indicating that its subsequent operation failed. Upon receiving such feedback, GenCache-feedback deletes the faulty cache and re-initiates the cache generation process for the corresponding cluster.
>
> ## Answers to Other Concerns
> 1. **Baselines rely on relatively small-context datasets; it is unclear how the regex-driven program will scale to long, multi-paragraph inputs or to irreversible agent actions?**
>
> We agree with the reviewer that GenCache is effective when prompts exhibit three key characteristics: (1) **Structural Regularity:** prompts follow a consistent format, e.g., the item to purchase always appear between fixed phrases like “buy” and “under the price of X dollars.” (2) **Controlled Variability:** while the structure stays the same, elements like item names, fault regions in system alerts, or optional words like “please” can vary. (3) **Predictable Response Patterns:** the responses generated follow a consistent format, even if they are not identical. GenCache accelerates agents handling repetitive tasks by leveraging this regularity.
>
> Importantly, GenCache *does not limit prompt or response length*. While short, human-written prompts often exhibit the above characteristics, longer multi-paragraph automated prompts (e.g., generated by system monitors) can also show similar characteristics.
>
> Currently, most humans interact with agents through a chatbot interface where prompts have high diversity. While GenCache may not be suitable for free-form chatbot interactions, we envision that LLM will be integrated into automated workflows where tasks will be repetitive. GenCache is *highly effective* for such use cases.
>
> For irreversible actions, mistakes can impact the underlying system. Although GenCache-feedback achieves the lowest negative hit rate, we recommend using it only when prompts are fully structured, differing in only attributes. As shown in Section 4.3, multi-step workflows can tolerate some errors, as LLM-generated rationales in subsequent steps may correct errors from earlier steps caused by negative hits.
>
> 2. **CodeGenLLM and ValidLLM both run on GPT-4o/4, which may be costly; demonstrating that a smaller, open model can handle synthesis and validation would strengthen practicality.**
>
> We will definitely consider running GenCache with smaller models like GPT-4.1 mini, GPT-4.1 nano, GPT-3.5, Llama, etc. by replacing the LLM API call.

---

> ### Author Response · Authors · 2025-08-07
>
> Thank you again for your thoughtful reviews. We wanted to check if there is any additional information or clarification we can provide alongside the rebuttal to support your evaluation of the paper. We would be happy to elaborate further on any aspect of our rebuttal or the paper itself.

---

> > ### Comment · Reviewer_znMf · 2025-08-07
> >
> > Thank you for the detailed responses. My concerns have been addressed. I maintain my accept rating.

---

### Official Review · Reviewer_Wqw4 · 2025-07-03

**Clarity:** 3
**Significance:** 4
**Originality:** 4
**Rating:** 5
**Confidence:** 4

**Summary:**

The authors have introduced GenCache, a caching mechanism designed to reduce the number of LLM calls, say in agentic workflows, by caching structurally similar prompts and symbolic cache programs which generates variation-aware responses on cache hits. The core insight is that in specific usecases, structural similarity in prompts often correlates with predictable variations in their corresponding responses. To exploit this, GenCache uses additional LLM calls to identify consistent transformation patterns between prompts and responses within each cluster and learn a transformation program. These learned programs act as dynamic response generators, improving over ExactCache and GPTCache like methods which fail to capture subtle variations that demand different responses.

**Questions:**

- See Weaknesses Section
- How are you selecting the exemplars from the cluster to prompt the CodeGenLLM for generating the cache program?
- While the authors acknowledge that the use case of GenCache is limited to agentic workflows where structural similarities often occur between prompts and response, it is unclear how the proposed caching would perform in agentic use-cases with dynamic workloads where the responses while being structurally similar are data dependent. Imagine a similar use case from Figure 1 but now the plan to look up AAA batteries/USB-C cables on Amazon needs to incorporate specifying min-max cost limit which is user-dependent and must be determined via an external database of user’s expenditure budgets. Can your program learning incorporate such database retrieval or tool call to collect external information?

**Ethical Concerns:**

["NO or VERY MINOR ethics concerns only"]

**Final Justification:**

The paper proposes a caching mechanism better than state of the art which would be a valuable addition for the research in this field. The authors have adequately addressed my concerns and based on the discussion, I recommend the paper be accepted.

**Limitations:**

Yes

**Quality:**

3

**Strengths And Weaknesses:**

**Strengths**

- The paper shows a clear motivation for GenCache by comparing with the limitations of existing Semantic and Exact caching mechanisms.
- GenCache is a promising direction in better weighing the trade-off between LLM-call cost optimisation and accuracy with caching.
- The structurally similar assumption is showcased and validated in real-life use cases of agentic workflows., where saving on LLM-call is demonstrated with GenCache.
- The authors have also showcased the limitations of GenCache with experiments when diverse structures start to appear in prompts during caching and the mechanism experiences lower cache hits, laying out the directions for future research and improvements.


**Weaknesses**
- Missing ablation with respect to similarity threshold, minimum prompts per cluster used to create clusters and corresponding cached programs. For instance, experiments to show how quickly each cached program converges and what happens when clusters contain outliers are missing.

- No sufficient reasoning/evidence to support design decisions during program validation.
    - What is the reasoning behind choosing 50% as the threshold on matched program-generated responses with exemplars? What happens when mismatched cases are mostly due to outliers?
    - Alternately, what does the mismatch between program generated responses and corresponding exemplars during program validation actually signify? Does it mean that the learned program is less generalisable or there are outliers present? Consequently, how do you make sure that while retrying program generation with reflection, you are not overfitting to outliers and generalising well?

---

> ### Author Rebuttal · Authors · 2025-07-31
>
> Thank you for your thoughtful suggestions and insightful reviews. Here are our responses to the questions.
>
> 1. **Missing ablation with respect to similarity threshold, experiments to show how quickly each cached program converges**
>
> **(Ablation 1) Similarity Threshold for Clustering**
> | $T^p$          | $T^r$                | Hit %  |
> | :---------------:| :------------------: | :------: |
> |  0.75            | 0.7                   | 79.72  |
> | 0.8 (default) | 0.75 (default)   | 83.66  |
> |  0.85            | 0.85                 | 60.02  |
>
> Thank you for the valuable suggestions. We conducted an ablation study on the similarity thresholds ($T^p$ and $T^r$ in Section 3.2), which determine the most similar cluster for a given prompt. A new prompt is added to the cluster if the cosine similarity with the cluster centroid crosses the above thresholds. In Sections 4.1 and 4.2, we use default values of $T^p = 0.8$ and $T^r = 0.75$. For this ablation, we varied the thresholds--one lower, one higher.
>
> We observed that with the higher thresholds, cache hit rate reduced significantly. Higher thresholds led to fewer prompts being grouped together, resulting in many small clusters that lacked the required number of exemplars ($\nu$) to generate a program. For a new prompt, it often resulted in being mapped to a cluster where there were no cached prompts. Even when a cached program existed, the program failed to extract the item name using regex due to overfitting on limited exemplars.
>
> Conversely, lowering the thresholds grouped more varied prompts (despite similar structure) into a single cluster. This variation made it difficult for CodeGenLLM to learn a consistent pattern, again reducing cache hits.
>
> As suggested, we plan to include a more extensive sensitivity analysis of $T^p$ and $T^r$ in the final version of the paper.
>
> **(Ablation 2) Quick Convergence of each cached program**
> Table 2 illustrates how quickly each cached program converges, i.e., how quickly a reusable program is created, against varying minimum numbers of exemplars. We see that as we increase the number of exemplars, it consistently reduces the number of the LLM calls required to generate a valid program. This is because the CodeGenLLM can infer the pattern better with more examples and reliably write a regex pattern that can work for a variety of prompts. This is referenced in Line 304. (We noticed a typo in the current version that incorrectly refers to Figure 2 instead of Table 2. We apologize for the oversight and will correct it.)
>
> 2. **What is the reasoning behind choosing 50% as the threshold on matched program-generated responses with exemplars?**
>
> We run a sensitivity analysis by varying the threshold for the proportion of exemplars that must match the program-generated response. We choose three threshold values--40%, 50% (default), and 70%, and run GenCache on Param-w-Synonym dataset (1000 user prompts) with the same setting as described for Table 1.
>
> | Threshold | Hit % | +ve Hit |
> | :-----: | :----: | :------: |
> |40% | 80.53 | 91.74 |
> |50% | 83.66 | 92.16 |
> |70% | 89.8 | 92.47 |
> |90% | 94.8 | 94.74 |
>
> We see that increasing the threshold indeed increases the cache hit rate. This indicates that with a lower threshold, the CodeGenLLM was able to create a program, but was not general enough to cater to all the minor variations in the prompt, as the reviewer correctly suggested. As the threshold increased, the CodeGenLLM was able to create a reliable program by considering more patterns in the regex and making the program general for more prompts. At 50% threshold, the generated program may have resulted in more cache misses than at 70% or 90%, but the positive hit rate remains similar. Thus, even though the recall is lower, the precision remains high. However, we observed that with higher thresholds, more LLM calls are required early in the run, when the database contains only a few prompts, to create reusable caches. Once sufficient prompts are clustered and cached programs are generated, cache misses become rare.
>
> We speculate that this sensitivity must also vary with $\nu$, since $\nu$ controls the total number of in-context examples used. With more in-context examples, it may be hard to create a reliable cache when the threshold is high.
>
> As per suggestion, we intend to do a complete sensitivity analysis with more runs in the final version of the paper.
>
> 3. **What does the mismatch between program generated responses and corresponding exemplars during program validation actually signify?**
>
> The primary reason for the mismatch between program-generated response and the corresponding exemplar response is **formatting differences** between the responses.  LLM responses (R) for prompts (P) are structured as **JSON**, not plaintext, because the agents expect a specific format of responses from the LLM. Thus, the formatting differences arise when the exemplar responses are structured as JSON, but program-generated responses are in plain text or they may lack specific key-value pairs present in the exemplars. This resulted in ValidLLM flagging the program-generated response as incorrect.
>
> The other reason, in certain cases, is that the LLM appeared to overfit to few exemplars from the list of exemplars provided during cache generation. As a result, responses aligned well with those few, but failed to generalize over the rest of the exemplars. This leads to regex mismatches and giving errors like “keyword not found” or only a subtring being extracted due to a more strict regex which does not correctly reflect the ground-truth exemplar response.
>
> 4. **How do you make sure that while retrying program generation with reflection, you are not overfitting to outliers and generalising well?**
>
> The reviewer has correctly pointed out that in some cases, the cached program might overfit to outliers, when the generated program relies on regex patterns that are overly tailored to a few examples (e.g., containing hardcoded keywords) rather than reflecting a generalizable structure.We tackle this in two ways. First, we prompt CodeGenLLM to include exception handling blocks at all steps in the program. If any step fails (e.g., regex does not match with the prompt), the program will return a specific keyword such as “None”. GenCache detects this failure and automatically falls back to the LLM to generate the response.
>
> Second, after identifying the most similar cluster for a given prompt, we also do a regex check on the prompt structure itself before using the cache. If the incoming prompt structure does not match the prompt structure for which the code was generated, GenCache bypasses the cache and again defaults to the LLM for response generation.
>
> 5. **How are you selecting the exemplars from the cluster to prompt the CodeGenLLM for generating the cache program?**
>
> CodeGenLLM is currently prompted using all exemplars within a cluster. For each cluster, we limit its size to 3$\nu$ in the database, where $\nu$ is the minimum number of exemplars that must be present in the cluster in order for CodeGenLLM to start generating a program. This constraint ensures that the number of in-context examples used in the prompt for CodeGenLLM is capped by 3$\nu$.
>
> Furthermore, to prevent the prompt-response pair within a cluster from increasing indefinitely, we append a user prompt-response pair to the most similar cluster in the database only when there is a cache miss and LLM generates the response instead of the cache.
>
> 6. **Imagine a similar use case from Figure 1 but now the plan to look up AAA batteries/USB-C cables on Amazon needs to incorporate specifying min-max cost limit which is user-dependent and must be determined via an external database of user’s expenditure budgets. Can your program learning incorporate such database retrieval or tool call to collect external information?**
>
> Thank you for highlighting this interesting use case. We clarify that GenCache is designed as an API layer that the agent queries by providing a fully-formed prompt, and it returns a corresponding response. GenCache internally decides whether to use a cache or query an LLM for the response.
>
> Importantly, GenCache’s scope is to use the complete prompt as received from the agent. This prompt is expected to include all the contextual and user-specific information required by an LLM to generate a response. GenCache uses this prompt-response pair to generate a program that can be reused for cache hit. The **agent is responsible for constructing the full prompt**, including any logic to retrieve user profiles from a database, injecting guardrails, or adapting the prompt based on context.
>
> Thus, GenCache acts as a caching layer that operates on the final prompt constructed by the agent. We believe that any database interaction or personalization logic falls within the agent's domain and not within the responsibilities of GenCache.

---

> > ### Comment · Reviewer_Wqw4 · 2025-08-05
> >
> > The authors have adequately addressed my main concerns in the rebuttal. The paper presents a valuable contribution to advancing techniques for LLM caching. I will maintain my original score.

---

### Note · Authors · 2025-08-13

We sincerely thank the reviewers for their thoughtful feedback.

**Key Contribution:** We introduce GenCache, a novel caching technique designed to accelerate inference while reducing LLM calls in repetitive agentic workflows. Unlike traditional caches that return stored responses verbatim, GenCache produces variation-aware outputs tailored to structurally similar prompts. It overcomes the limitations of exact and semantic caches, which struggle to support prompts with small but meaningful variations, common in agentic workflows.

**Reviewer Consensus & Strengths:** We’re pleased that all reviewers rated the work positively, highlighting its novelty, utility, technical depth, and evaluation. Their comments were insightful. They raised important questions, which our rebuttal addressed as follows:

*Reviewer Wqw4:* We added ablation studies on clustering thresholds and varied the ratio threshold for cache validity based on program-generated responses. We clarified how we control overfitting to outliers, select exemplars, and what ValidLLM mismatch signifies. The reviewer was satisfied and maintained their score of 5.

*Reviewer znMf:* We measured CPU overhead for clustering and cache generation, tested different ratio thresholds for cache validity, clarified GenCache’s interpretation of LLM responses, and explained why performance remains stable despite more in-context exemplars. We showed that effectiveness depends on structural regularity and predictable patterns, not prompt length. The reviewer was satisfied and maintained their score of 4.

*Reviewer 7qKt:* We detailed computational overheads for cache lookup, database insertion, and vector computation. We clarified the experimental setup, dataset creation, and evaluation metrics with examples. We however noted its limited applicability to free-form text. The reviewer was satisfied and raised their score.

*Reviewer 6e2p:* We discussed GenCache’s scope, emphasizing its suitability for repetitive agentic workflows but not for free-form chat. We also mentioned plans to demonstrate GenCache’s generality using an additional dataset. The reviewer increased their confidence and maintained their score of 4.

**Final Remarks:** All reviewers agree that GenCache is a novel and practical caching solution for LLM prompts. They are happy with the technical contribution and are satisfied with our rebuttal. We plan to incorporate their suggestions and new results into the final version to further strengthen the work.

---

### Decision · Program_Chairs · 2025-09-17

**Decision:**

Accept (poster)

**Comment:**

This paper introduces a generative caching mechanism for LMs that targets structurally similar prompts. The primary contribution is a novel 'cache-the-program' approach where, instead of storing static responses, the system caches small, executable programs that generate variation aware outputs, improving hit rates and reducing latency in repetitive, agentic workflows.

All four reviewers are in agreement on the paper's acceptance, identifying its key strengths as the novelty of the generative caching approach, its practical utility for agentic workflows, and its strong empirical results. The initial reviews raised several points for clarification, questioning the method's limited scope beyond structured prompts, the lack of detail on computational overhead, and the absence of ablation studies for key hyperparameters. The reviewers were satisfied with the authors' detailed response, which provided new results and clarified the method's intended application.

For the camera-ready version, I recommend that the authors integrate the detailed clarifications and new results from their rebuttal into the main body of the paper. This includes: (1) the new ablation studies on key hyperparameters like clustering and validation thresholds, (2) the detailed breakdown of computational overheads for cache lookup and generation, and (3) the expanded discussion on the method's scope, clearly positioning it for structured, repetitive workflows rather than general-purpose chat.